# Digital economy development and global value chain network centralization

**Hankun Yuan[1], Mingyang Yue[2]*, Rong Wang[3], Biao Ren[2], Xueyang Li[2]**

**1** Department of World Economy and Politics, Party School of C.P.C Jiangsu Provincial Committee, Nanjing, Jiangsu, China, **2** Department of Economics, Party School of C.P.C Jiangsu Provincial Committee, Nanjing, Jiangsu, China, **3** Wuhan University, Wuhan, Hubei, China

* yuemingyang93@163.com

## Abstract

The development of the digital economy significantly relies on promoting economic liberalization and achieving high-end embedding of manufacturing into the global value chain (GVC). Based on the heterogeneous firm model, this paper introduces the factors of digital economy development, aggregates the production behavior of firms at the national level, and empirically examines the effects of digital economy development on the centralization of the GVC network and the internal mechanisms. The results show that the development of the digital economy significantly enhances a country's centralization of GVC network, primarily through enhancing productivity and alleviating resource misallocation. Meanwhile, digital economy development contributes more to the centralization of the GVC network in developed countries and high institutional quality industries. Further analysis shows that digital economy development not only enhances a country's centralization of GVC network but also extends the length of a country's GVC and promotes a country to climb up the GVC.

## 1. Introduction

In recent years, the digital economy, represented by the internet, big data, cloud computing and blockchain technology, has increasingly emerged and continues to integrate into all aspects of social development. According to the "China Digital Economy Development Report (2022)" released by the China Academy of Information and Communication Research, the value added of the digital economy in the world's main 47 major countries reached 38.1 trillion yuan in 2021, accounting for 45% of the total global GDP. China's digital economy reached $7.1 trillion, accounting for more than 18% of the total of the 47 countries, ranking second in the world. Enterprises and industries, as the main carriers of economic activities, are bound to be affected by the development of the digital economy in their production activities, and the connection between the digital economy and real industries is becoming increasingly closer. In this context, countries have introduced policies to support the development of the









**Data availability statement:** WIOD database: https://www.rug.nl/ggdc/valuechain/wiod/. UIBE GVC Indicators database: http://gvcdb.uibe.edu.cn/. All data are fully available without restriction.

**Funding:** This study was funded by the National Party Schools of C.P.C. (Academy of Governance) System Principle Research Project (Grant Number: 2023DXXTZDDYKT032), and the Jiangsu Office of Philosophy and Social Science (Grant Number: 23EYC019).

**Competing interests:** The authors have declared that no competing interests exist.

digital economy to realize the transformation and upgrading of the manufacturing industry, such as "Made in China 2025" in China, "German Industry 4.0" in Germany and "National Robotics Initiative 2.0" in the United States. All these strategic policies regard the digital economy as the top priority for national competition. It is observed that enhancing the competitiveness of countries in global value chains requires relying on the support of high technology represented by the digital economy; this provides a rich practical significance and realistic background for the research of this paper.

In addition, the digital economy plays an important role in the field of foreign trade [1]. Currently, given the further development of the GVC network, decentralized and fragmented production methods have become mainstream, and countries can undertake different production links to jointly complete the production of products. Taking China as an example, since joining the WTO in 2001, China has undertaken a large amount of processing and assembly work in the production of GVC with the advantage of cheap labor, has achieved rapid development of manufacturing while deeply integrating into the trade network of GVC, and has surpassed developed countries such as Europe and the United States in terms of ranking first in the center of the GVC network [2]. The global value chain network not only reflects the overall world trade pattern and trade connections but also demonstrates the importance of countries in global trade and whether they are in pivotal positions. If a country is positioned more centrally in the trade network, it signifies a stronger ability to control and access resources and information. Whereas the realization of GVC network centralization is a manifestation of countries' competitiveness in the GVC system; unfortunately, most studies currently focus on the drive of the digital economy for upgrading the GVC and have not yet deeply examined the impact of the digital economy on the centralization of the GVC network. Then, the key questions of this research are as follows: with the development of the global digital economy, what is its impact on the position of countries in the global value chain network? What kind of mechanism of action drives this impact? The answers to these questions are conducive to promoting the high-quality development of the manufacturing industry and realizing a higher level of integration into the global value chain network.

In terms of theoretical research, this paper proves that the development of the digital economy promotes the centrality of the global value chain network by constructing a heterogeneous enterprise model, and that this promotion is mainly realized through the improvement of productivity and the reduction of resource allocation efficiency. With the development of the digital economy, it creates advantages for enterprises to participate in global value chain competition, and therefore promotes the country's production and cooperation in the global value chain network, and promotes its gradual development towards the center of the global value chain network.

This paper explores the impact of the development of the digital economy on the centralization of GVC network mainly through theory and empirical evidence, and we find that, first, the digital economy significantly enhances the status of countries in GVC network. Second, this enhancing effect varies with the degree of economic development and institutional environment of each country. Third, the development of the digital

economy achieves the centralization of GVC network through productivity enhancement and resource mismatch mitigation. Fourth, the digital economy effectively enhances a country's breadth and depth in global value chains. In addition, there is room to expand the research in this paper, such as considering the impact of GVC network centrality from the firm level.

Despite the growing body of literature examining the impacts of the digital economy on economic growth, labor markets, and trade patterns, significant research gaps remain regarding its direct and indirect effects on the centralization of global value chain (GVC) networks and the underlying mechanisms driving these changes. While existing studies have extensively explored how the digital economy facilitates the upgrading of GVC, mostly studies have largely overlooked its influence on the centralization of GVC network, which reflects a country's pivotal role and control over resources and information flows in the global economy. In recent years, the digital economy, fueled by advancements in information and communication technologies (ICTs) such as the internet, big data analytics, cloud computing, and blockchain, has emerged as a transformative force reshaping the global economic landscape. This digital revolution is not merely confined to technological innovation; it is fundamentally altering the way businesses operate, goods and services are produced, and value is created and distributed across borders.

In summary, this paper contributes to the existing literature by providing a nuanced understanding of the digital economy's role in shaping the centralization of GVC network. By integrating firm heterogeneity and examining both direct and indirect effects, we offer a comprehensive framework that elucidates the complex dynamics between digital economy development and GVC network centralization. Our research not only advances theoretical knowledge but also provides actionable insights for policy-makers aiming to leverage the digital economy for economic growth and global competitiveness.

## 2. Literature review

The research related to this paper focuses on the following aspects: Existing research has explored the impact of the digital economy from multiple perspectives, most of which focus on the impact on labor force employment. First, the digital economy represented by AI has a strong substitution effect on the labor force particularly in traditional labor-intensive industries, inevitably leading to an increase in unemployment [3–4] and a decrease in the share of labor [5], but it also has a substitution effect on the traditional labor force in terms of creating new jobs [6]. Second, the digital economy has a significant boost to overall productivity, as it has strong technology-intensive attributes and a strong spillover effect compared to traditional industries, and hence it can have a significant boost to industry-wide productivity [7]. Finally, given the penetration of the digital economy, enterprises face digital transformation, and their production and operation activities are inevitably affected by the digital economy [8–9]. Most of the studies mentioned above have focused on the impact of the digital economy on the "binary margin" of firms' exports, but few studies have focused on its impact on the quality of export products [10]. Given the continuous penetration of the digital economy and the internet, Ma & Hu explored the impact of the digital economy on multiproduct exporters and found that information technology prices have been decreasing, reducing firms' export costs and improving the quality of their export products while also allowing them to spend more resources on producing their core products to maintain their competitiveness in the international market [11]. Additionally, research suggests that information and communication technology (ICT) plays a significant role in driving global value chain (GVC) activities [12]. Studies from both macro and micro perspectives have found that the development of the internet notably promotes GVC trade [13]. This promotion is mainly achieved by enhancing supply-demand matching and fostering communication and collaboration, injecting new vitality into the global value chain [14]. However, there are also opinions suggesting that ICT may have adverse effects on the upgrading of GVCs. Under the global value chain division system driven by labor arbitrage, the substitution effect of automation on labor leads to the contraction of GVC trade [15].

At the same time, there is a large body of literature using social network analysis as a research tool to analyze and measure the intricate trade issues among countries. The advantage of social network analysis is that it can reflect not only the overall trade pattern in the world but it can also reflect the importance and centralization of a country and each

industry in the global trade network. If a country is at the center of the trade network, its access to resources and profit is stronger. Traditional social network analysis usually considers each country as a point in the network, and the line linking the points is the bilateral trade flow. The analysis of the global trade network reveals that trade and economic exchange between countries are becoming more frequent [16–17]. Meanwhile, global trade networks are not static. Akerman & Sein studied the change in the global trade network in 1995–2011 and found that it is moving toward intensification, clustering, and decentralization [18]. In their analysis of the global trade network, Amighini & Gorgoni found that developing countries are playing an increasingly important role in global trade and that automated production is having an impact on previous trade patterns [19]. The above studies analyze the change in the global trade network mainly through bilateral trade volumes. However, with the changing division of labor systems in GVC, intermediate goods are often traded across multiple borders, so the value of a country's final export may originate from many other countries. Traditional social network analysis, which mainly uses total trade, has led to a misunderstanding of today's global trade pattern, so the GVC network should be reinterpreted using value-added trade [20]. To overcome the shortcomings of traditional trade statistics, value added flows are used as "edges" in trade networks, and thus, the construction of a global trade network is more in line with the characteristics of today's GVC trade [21].

The change in firms in terms of the division of production in GVC has been measured using the concept of the number of production stages [22]. Currently, the division of labor in GVC is facing many challenges, and COVID-19 has had a great impact on the global production supply chain [23–25], as it slowed the growth of the global economy and gradually reduced the incentive of countries to participate in the division of labor in GVC [26]. The rise of artificial intelligence (AI) has had an impact on GVC, as the application of AI increased the efficiency of using intermediate goods to produce final goods, making the production process more automated and smarter, saving labor costs [27], and improving the division of labor of firms in GVC.

Most studies currently only use traditional trade statistics methods to study global trade networks or focus only on the specifics of GVC network, lacking the exploration of the influencing factors of GVC network, and there is even less literature studying the role played by the digital economy in this literature. This paper combines the advantages of trade networks based on traditional trade measurements and GVC value-added measurements to construct a trade network reflecting the development of GVC in specific countries; the research in this paper is another breakthrough for the social network analysis method in the study of global value chains and it bridges the gap in this field.

In contrast to the current literature, this paper investigates the impact of digital economy development on the centralization of GVC network from theoretical and empirical aspects and makes innovations in the following three aspects.

First, this paper extends the heterogeneous firm trade model, introduces digital economy development factors in firms' production behavior and export decisions, and gauges firm behavior at the country level to construct a theoretical framework of the impact of digital economy development on the centralization of GVC network.

Second, this paper uses country-industry level data to empirically study how digital economy development affects the centralization of the GVC network. Referring to Arnold et al. and Xiao et al., this paper uses the UIBE (University of International Business and Economics) GVC Indicators database and the WIOD database to construct country-industry level digital economy development and GVC network centralization indicators [28–29]. Considering possible endogeneity problems, this paper refers to Nunn & Qian and uses the number of landline telephones in each country in 1984 as an instrumental variable for the least squares analysis [30].

Third, this paper summarizes two major channels through which the development of the digital economy affects the centralization of the GVC network, namely, the productivity channel and the resource allocation channel. Numerous studies have shown that digital economic development effectively reduces export fixed costs, while the digital economy has a significant substitution effect on labor and it increases labor productivity. With the increase in productivity, a country has a more competitive advantage in the GVC network. In addition, the development of the digital economy facilitates the free flow of production factors among national sectors, alleviating resource mismatch, enabling advantage gains for a country in global competition, and promoting such a country to the center of the GVC network.

## 3. Theoretical model and research hypothesis

In this paper, based on the Melitz & Ottaviano model [31], we construct a heterogeneous firm export decision model and analyze its impact on GVC network centralization by introducing the factors of digital economy development. According to the definition of Xiao et al. and Amador & Cabral, the centralization of the GVC network is mainly reflected in two aspects: the number of trading partners and the value added of exports. Therefore, the theoretical model in this paper starts from these two aspects and aggregates firms' export decisions at the national level and then it analyzes the impact of digital economic development on the centralization of the GVC network.

In the theoretical model, we have retained the basic framework of Caliendo & Parro [32]and extended the endogenous growth model of De Ridder [33]to construct a theoretical framework for corporate export behavior incorporating data as a factor of production.

### 3.1 Consumer preference and demand

Drawing on the setting of Melitz & Ottaviano, this paper defines the consumer utility function as a proposed linear utility function:

$$U = q_0 + \alpha \int_{i\in\Omega} q_i di - \frac{1}{2}\gamma \int_{i\in\Omega} (q_i)^2 di - \frac{1}{2}\eta \left(\int_{i\in\Omega} q_i di\right)^2 \tag{1}$$

where $q_0$ represents a homogeneous product (also called a valued good) with a constant price of 1; $q_i$ represents a heterogeneous product; $\gamma$ represents the degree of product differentiation between the varieties; $\alpha$ and $\eta$ represent the elasticity of substitution between homogeneous and heterogeneous products; and $\Omega$ is a set of indexes. The inverse demand function can be derived under the utility maximization condition as follows:

$$p_i = \alpha - \gamma q_i - \eta Q \tag{2}$$

Among them, $Q = \int_{i\in\Omega} q_i di$. In addition, the linear market demand function can be written according to Equation (2) as:

$$Lq_i = \frac{\alpha L}{\eta N + \gamma} - \frac{L}{\gamma} p_i + \frac{\eta N}{\eta N + \gamma}\frac{L}{\gamma}\overline{p} \tag{3}$$

where$L$ represents market demand. $\overline{p} = (1/N^{I}) \int_{i\in\Omega} p_i di$. $N$ stands for the type of consumer goods and can be seen as the number of exporters. To ensure that the demand is always positive ($q_i > 0$), the maximum price for the firm can be obtained as:

$$c_D = p_{\max} \le \frac{1}{\eta N + \gamma}(\alpha\gamma + \eta N\overline{p}) \tag{4}$$

$c_D$ represents the maximum price, which can be considered the critical cost; that is, when the firm production cost is higher, the enterprise stops production. When it is lower, the firm continues to produce.

### 3.2 Producer behavior

In the behavior of producers, data elements in the digital economy are introduced as production factors into enterprises' production decisions, influencing their operational performance [34]. This further emphasizes the special status and role of the digital economy within the model.

Why can data within the digital economy serve as a production factor influencing corporate production decisions? The reason is that the digital economy relies on data as a key production factor, with digital technology as its core driving force and modern information networks as its important carrier. Through the deep integration of digital technology with the real economy, it continuously enhances the digitization, networking, and intelligence levels of the economy and society, accelerating the reconstruction of economic development and governance models. Data is a crucial production factor in the digital economy era, often compared to coal and oil in the digital age. Firstly, as a new input, data can drive economic growth on its own. Unlike traditional production factors such as capital and labor, data has characteristics such as extremely low search costs, zero replication costs, extremely low transmission costs, and non-excludability [35]. Data not only has nearly infinite supply but also has a wide range of applications, thus being able to break through the growth constraints of traditional production factors. Mainstream economic growth models have begun to introduce variables related to the digital economy, such as data capital and robots. Secondly, digital technology enhances capital and labor factors, increasing their marginal output. From a capital perspective, the diffusion of digital technology can reshape a firm's production and management methods, enhancing the efficiency of capital accumulation. Therefore, in the current digital economy era, data as a key factor is a fundamental characteristic of the digital economy, continuously influencing corporate production decisions.

In the producer model, we assume that firms require labor input ($l_i$) and data capital ($f_i$) to produce goods. Since big data enables firms to make more scientific production decisions, we assume that data capital investment can reduce firms' marginal costs ($s_i$). That is, $c_i = s_i\omega$, where $s_i \in [0, 1]$, $\omega$ stands for wage level Hence, the production function of the firm can be expressed as:

$$y_i = \frac{l_i}{s_i} \tag{5}$$

The combination of data capital and labor has two characteristics in terms of its impact on output: 1. There are differences in the efficiency of different firms in developing and utilizing data capital, $\varphi_i$ represented by firm-specific cost parameters. Due to the economies of scale and scope of data factors, cost parameters $\varphi_i$ significantly decrease with increased data capital investment $f_i$. 2. Firms invest in data capital before observing their competitors' marginal costs and pricing. When firms set prices and make production decisions, data assets have already become sunk costs.

The functional relationship between a firm's data capital $f_i$ and the decrease in marginal cost $s_i$ is given by $f(s_i, \varphi_i)$. The function $f_i$ is twice differentiable, strictly decreasing when $s_i \in [0, 1]$, and increasing when $\varphi_i > 0$. It is assumed that $f(s_i, \varphi_i)$ is convex and decreasing in $s_i$, and satisfies $f(1, \varphi_i) = 0$, $\lim_{s_i \to 0} f(s_i, \varphi_i) = \infty$. For the purpose of analysis, this paper selects an exponential function form for the data capital function that meets these characteristics:

$$f(s_i, \varphi_i) = \varphi_i \left( \left[ \frac{1}{s_i} \right]^{\psi} - 1 \right) \tag{6}$$

Where $\psi > 0$ represents the curvature parameter.

Subsequently, the decision-making process for the firm unfolds as follows: First, the firm decides on its data capital expenditure $f_i$, that is, whether to produce at a marginal cost lower than the benchmark $\omega$; then, the firm select $s_i$and incurs the corresponding data capital expenditure $f_i$; finally, the firm observes its competitors' marginal costs and sets its price $p_i$. The analysis will proceed using backward induction.

Decision on Data Capital Investment: Combining Equation (4), the $s_i$ that minimizes marginal cost at the optimal output $y_i^*$ is:

$$s_i^* = min \left[ \left( [y_i^*]^{-1} \psi \varphi_i \right)^{\frac{1}{\psi+1}}, 1 \right] \tag{7}$$

The modeling approach for producer behavior can be summarized as follows: With the development of the digital economy, data elements $f_i$ are invested as important production factors in enterprise production, and as the digital economy becomes more developed, the marginal cost $s_i$ of enterprises decreases.

Then, according to the profit maximization condition (MR = MC) and combining Equations (2) and (4), and under the equilibrium condition, the firm's output, price and profit can be expressed as:

$$q(c) = \frac{L}{2\gamma}(c_D - c) \tag{8}$$

$$p(c) = \frac{1}{2}(c_D + c) \tag{9}$$

$$\pi(c) = \frac{L}{4\gamma}(c_D - c)^2 \tag{10}$$

### 3.3 Equilibrium analysis

According to the definition of Xiao et al., the centralization of the GVC network is reflected in trading partners and export value added. First, the number of trading partners in the GVC network mainly reflects the economic and trade transactions between countries; the more trading partners a country has, the more the country is at the center of the GVC network. Therefore, the more exporters a country has, the more products the country exports to other countries, indicating that there are more global trading partners, hence the more the country moves to the center of the global value chain network.

In equilibrium analysis, the number of exporting firms in the market is determined by the profit equilibrium condition:

$$\int_0^{c_D} \pi(c)dG(c) = \frac{L}{4\gamma}\int_0^{c_D}(c_D - c)^2 dG(c) = fe \tag{11}$$

where $fe$ represents the fixed cost of exporters, namely, the fixed cost required by firms to enter the export market; $G(c)$ represents the distribution function of the marginal cost of firms, following the setting of the Melitz & Ottaviano model, which considers that the marginal cost of firms obeys the Pareto distribution with parameter $\kappa$, namely, $G(c) = (c/c_M)^{\kappa}$, and $c_M$ implies the upper limit of the marginal cost.

According to Equation (11), the number of exporters in the equilibrium condition can be obtained as:

$$N = \frac{2\gamma}{\eta}\frac{\alpha - c_D}{c_D - \overline{c}} \tag{12}$$

where $\overline{c} = \left[\int_0^{c_D} c dG(c)\right] / G(c_D)$ represents the average marginal cost of exporters in the market. Furthermore, the number of exporters under the equilibrium condition can be expressed as:

$$N = \frac{2(\kappa+1)}{\eta}\frac{\alpha - c_D}{c_D} \tag{13}$$

From Equation (13), it is clear that the number of exporters in the market is related to the critical cost and production cost of the firm ($\overline{c} = \frac{\kappa}{\kappa+1}c_D$). As the digital economy continues to develop and digital economy continues to penetrate into various industries, the decrease in marginal costs ($s_i$) for enterprises subsequently leads to a reduction in the average costs of enterprises in the market, that is, $\partial\overline{c}/\partial s > 0$. Thus, the relationship between the number of firms and the development of the digital economy can be expressed as:

$$\frac{\partial N}{\partial s} = \underbrace{\frac{\partial N}{\partial c_D}}_{-} \cdot \underbrace{\frac{\partial c_D}{\partial \overline{c}}}_{+} \cdot \underbrace{\frac{\partial \overline{c}}{\partial s}}_{+} < 0 \tag{14}$$

Equation (14) shows that as the digital economy develops, the firms' production costs decrease, which in turn increases the number of firms in the export market. The greater the number of exporting firms in a country, the more the country is at the center of the global value chain network.

As noted above, the development of the digital economy increases the number of exporters, and it is another manifestation of the centralization of the GVC network in terms of added value of exports. Referring to Kee & Tang, using of digital technology, the revenue of firms can be expressed as [36]:

$$PY = \pi + wl + rk + sf + P^D M^D + P^I M^I \tag{15}$$

where $P^D M^D$ represents domestic production of raw materials; $P^I M^I$ represents foreign import of raw materials, and a further decomposition of Equation (15) can be obtained:

$$DVA = \pi + wl + rk + sf + q^D + \delta^D \tag{16}$$

where $DVA$ represents the domestic value added of exports; $q^D$ represents the imported portion of materials used domestically; and $\delta^D$ represents the domestically produced portion of materials imported from abroad. From Equation (16), it is easy to observe that as more digital economy is invested, the domestic value added of a country's export increases, helping to enhance the centralization of a country in the GVC network.

Based on this, this paper suggests Hypothesis 1.

H1: The development of the digital economy can enhance the centralization of a country in the GVC network.

### 3.4 Mechanism analysis

The previous analysis shows that the development of the digital economy has an obvious role in promoting the centralization of a country's GVC network; thus, the channels through which the digital economy affects the centralization of a country's position are the focus of this section.

First, in the productivity channel, according to the assumption of the Melitz & Ottaviano model, productivity can be viewed as the inverse of the marginal cost of a firm $(1/c)$. Thus, with N firms in the equilibrium market, a country's total export earnings can be expressed as:

$$R = Np(c)q(c) = \frac{NL}{4\gamma}\left[(c_D)^2 - c^2\right] \tag{17}$$

With Equation (17), it is easy to find that when a country's digital economy continues to develop, the penetration of digital economy into the industry deepens, and the production cost of the firm continues to fall, bringing about an increase in firm productivity $(1/c)$. Aggregating at the national level, that is, the more export earnings there are, the more firms engage in exporting and eventually such firms realize the central position of a country in the GVC network.

This is followed by the resource allocation channel, where the development of artificial intelligence is an important vehicle for the development of the digital economy, as artificial intelligence has a significant substitution effect on the labor market [37]. Therefore, based on previous studies, this paper divides the labor force into the general labor force ($la$) and artificial intelligence ($ai$), that is, $l = la + ai$. Referring to Bai et al., resource mismatch is introduced into the firm export model, and labor distortion is defined as$\tau_L$ [38]. When $\tau_L > 1$, the labor distortion is greater. At this point, the cost faced by the firm is:

$$TC = w_{la}la + \tau_L w_{ai}ai + rk + sf \tag{18}$$

Current research suggests that general labor costs are higher than AI costs and that with the popularization of artificial intelligence applications, there is a significant reduction in general labor wages. This in turn alleviates resource mismatch,

reduces production costs, increases a country's overall export earnings, and thus enhances the central position of a country's GVC network. Therefore, two additional hypotheses are proposed in this paper.

H2: The digital economy contributes to the central position of a country in the GVC network by increasing productivity.

H3: The digital economy contributes to the central position of a country in the GVC network by mitigating the degree of resource mismatch.

Fig 1 illustrates the mechanism through which the development of the digital economy influences the centralization of global value chain networks. Specifically, the integration of data elements into firms' production activities enhances the centralization of global value chain networks by improving productivity and alleviating resource misallocation through these two channels.

## 4. Study design

### 4.1 Method selection and model setting

Making a comprehensive assessment, this paper chooses the panel fixed-effects model (FE) to test the impact of digital economy development on the centralization of global value chain networks. There are three main reasons for choosing this method. First, from the data point of view, this paper mainly uses cross-country panel data from 2004–2014 for the study, which is in line with the conditions for the use of panel models. Second, compared with ordinary least squares (OLS) and the panel random effects model, the panel fixed effects model considers individual and time effects, making the parameter estimation more accurate. Third, this paper further uses the Hausman test to determine that it is more reasonable to use the panel fixed effects model. Therefore, this paper uses the panel fixed effects model as the main method of empirical research.

Referring to the study of Huang & Song, to explore the impact of digital economy development on the centralization of the GVC network, this paper constructs a regression model controlling for country, industry, and time fixed effects in the following form:

$$centralization_{cit} = \beta_0 + \beta_1 digit_{cit} + \beta_i \sum control_{cit} + \theta_c + \gamma_i + \lambda_t + \gamma_i \lambda_t + \varepsilon_{cit} \quad (19)$$

where $centralization_{cit}$ represents the level of GVC network centralization in industry $i$ of country $c$ in year $t$, and the larger its value is, the more central the country is in the GVC network; $digit_{cit}$ represents the industry digital economy penetration rate in industry $i$ of country $c$ in year $t$; $control_{cit}$ represents a series of control variables; $\theta_c$ represents country fixed effects; $\gamma_i$ represents industry fixed effects; $\lambda_t$ represents time fixed effects; $\gamma_i \lambda_t$ represents industry and time interaction fixed effects; and $\varepsilon_{cit}$ represents random disturbance term.

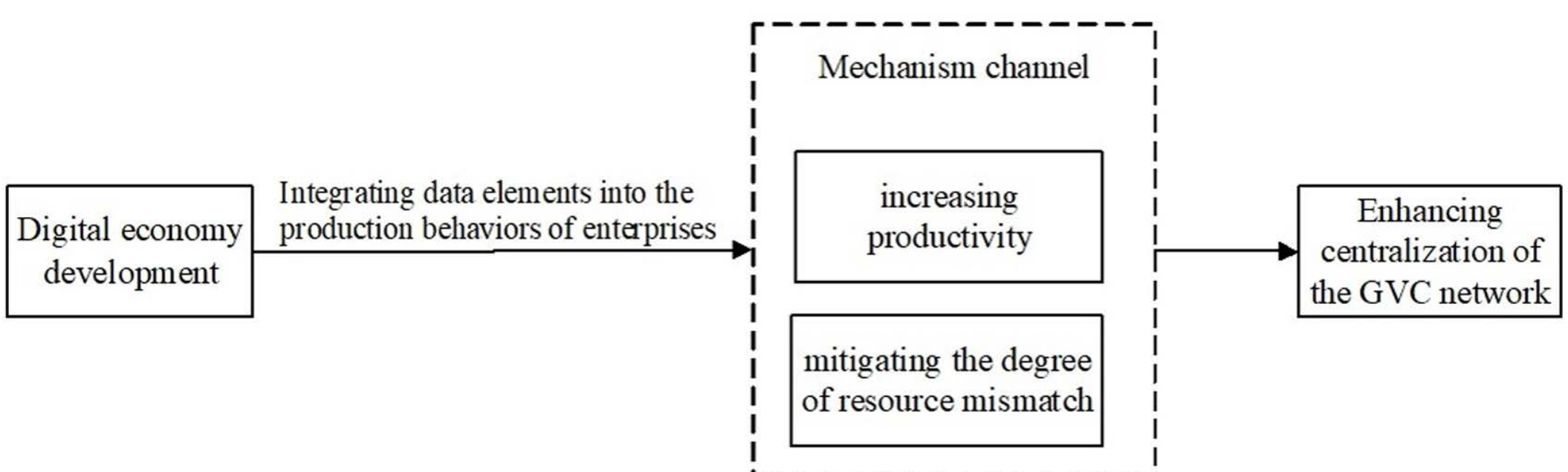


**Fig 1. A diagram illustrating the mechanism by which the development of the digital economy influences the centralization of global value chain network.**


### 4.2 Calculation of main variables

**4.2.1 Explained variable: GVC centralization.** The explained variable in this paper is GVC centralization, which has the advantages of both trade network centralization and GVC production division of labor measurement and can reflect a country's position more comprehensively and accurately. The traditional trade network index is mainly calculated by the traditional import and export value, which cannot reflect the characteristics of present GVC trade. The WWZ method proposed by Wang et al. decomposes the trade flows of each country, creating a basis for calculating the position of each country in the GVC network. Therefore, this paper refers to Amador & Cabral and Xiao et al. to calculate the centralization of a country using the domestic value added of each country's export as the trade flow between the network of each country. The specific GVC network centralization indicator is measured in three steps:

Step 1: Decomposition of value added of trade flow for each country. Referring to Wang et al., bilateral trade flows are decomposed at the country-industry level into four components: foreign value added (FVA), domestic value added (DVA), returned domestic value added (RDV_G), and pure double counting (PDC). To reflect the trade relationship between countries in GVC, this paper uses domestic value added (DVA) as the "connecting edge" of the trade network between two countries. The economic meaning of DVA is the domestic value added of exports from one country to another.

Step 2: Construction of the trade network. The GVC network consists of nodes and edges, with the edges being the domestic value added (DVA) in the trade of each country mentioned above and the nodes being the countries, where $V_c = [v_c]\ (c = 1, 2, \cdots n)$ represents the starting country, $V_j = [v_j]\ (j = 1, 2, \cdots n)$ represents the destination country, and $n$ represents the number of countries. In addition, this paper uses the weight matrix $W = \left[w_{cj}^{nt}\right]$ to denote the size of domestic value added in trade between countries, where $w_{cj}^{nt}$ represents the domestic value added from industry $n$ in country $c$ to country $j$ in year $t$. Thus, $V_c V_j$ and $W$ constitute the complete global value chain network.

Step 3: GVC network centralization construction. According to Amador & Cabral, the GVC network centralization indicator can be written as $centralization_{cit} = \sum_j w_{cj}^{nt}$; the larger the value, the more domestic value added a country's industry exports to other countries and the more the country is at the center of the GVC network. It is a better indicator of the number of trading partners and the intensity of a country's value added trade than a trade network based on total exports.

**4.2.2 Core explanatory variable: digital economy development.** Referring to Fernandes et al., the core explanatory variable is digital economic development, using industry digital economy penetration as an alternative variable for digital economic development. The indicator is composed of three components as follows:

(1) The level of digital development of the country. Referring to Ma & Hu, the national digital development level is constructed from three dimensions: digital infrastructure, digital technology innovation and digital competition intensity, and this paper compiles data from 39 major economies around the world from 2004–2014 and derives the level of digital economy development under the three dimensions of each country by principal component analysis.

(2) Industry digitization rate. Referring to Calvino et al., industries in the WIOD were classified according to ICT industrial policy, robot utilization, and other indicators, and "medium-high digital intensity industries" were selected as C7-9, C17-23, C28-30, C37-43, C45-50 and C54 (the above industries are all industry codes in WIOD). Then, using the total input weight of industries in the "medium-high digital intensity industries" and using the full coefficient consumption method, we measured the digitization ratio of industries in the GVC system as follows [39]:

$$ratio_{ij} = a_{ij} + \sum_{k=1}^{n} a_{ik}a_{kj} + \sum_{s=1}^{n}\sum_{k=1}^{n} a_{is}a_{sk}a_{kj} + \cdots \tag{20}$$

where $ratio_{ij}$ represents the digitization rate of the industry; $a_{ij}$ represents the direct consumption coefficient of industry $i$ in the "medium and high digital intensity industry"; the subsequent items are the indirect consumption of the previous items, and the cumulative sum is obtained as the complete consumption coefficient.

(3) Industry digital economy penetration rate

Considering the heterogeneity of industry development in each country, the degree of reliance on the digital economy varies from industry to industry. Even for the same industry, the difference in development of the digital economy in each country will have an impact on industry digital penetration. Therefore, referring to Arnold et al., the country-industry level digital economy penetration rates are constructed as follows [29]:

$$digit_{cit} = \sum_j DEI_n_{ct} \times ratio_{ij} \tag{21}$$

where $digit_{cit}$ represents the digital economy penetration rate in industry $i$ of country $c$ in year $t$; $ratio_{ij}$ represents the penetration of industry $i$ by the "medium and high digital intensity industry $j$"; and $DEI_n_{ct}$ represents the digital economy development level of country $c$ in year $t$.

**4.2.3 Other control variables.** The control variables in this paper mainly include (1) the labor force ratio (Labor), expressed as the share of the labor force population aged 15–64 in the total population. (2) Population density (Persons), expressed as the logarithm of the national population. (3) Industry size (Scale), expressed as the logarithm of total industry output. (4) Foreign direct investment (Fdi), expressed as the proportion of foreign direct investment in the country's GDP. (5) GDP growth rate (GDP), expressed as the deflated GDP growth rate. (6) Number of regional trade agreements (RTAs), expressed as the number of RTAs signed by the country.

### 4.3 Data description and descriptive statistics

**4.3.1 Data description.** This paper uses combined country-industry data from 2004–2014 to examine the impact of digital economic development on the centralization of a country's GVC network. The data are mainly obtained from the following three databases: the WIOD database, the UIBE GVC Indicators database, and the World Development Indicators database (WDI, World Developing Indicators). The global value chain network centralization indicator is obtained from the UIBE GVC Indicators database and it decomposes the bilateral trade economy using the WIOD 2016 edition; the industry digital economy penetration data are obtained from the WIOD 2016 edition; and other control variables are obtained from the WDI database.

(1) WIOD database

This paper uses the 2016 version of the WIOD database, including country-industry level input–output data from 2004–2014, to calculate the core explanatory variables in the paper [40]. We also acknowledge that the current WIOD database is only updated up to 2014, which poses certain limitations to the research in this paper. If the WIOD database is updated in the future, further research can be conducted based on this paper.

(2) UIBE GVC Indicators database

The UIBE GVC Indicators Database is an open-access database constructed and maintained by the Global Value Chain Institute at the University of International Business and Economics (UIBE). This database is designed to provide useful indicators and data support for international trade research and global value chain (GVC) analysis.The UIBE GVC Indicators database is based on the input–output data of countries and regions in the world and utilizes the structural decomposition method of input–output to calculate the indicators of value-added trade and GVC participation, which are used to calculate the core explanatory variables of this paper.

(3) World Developing Indicators Database

The World Developing Indicators database is constructed by the World Bank, and it includes annual economic development indicators of more than 200 major countries or regions around the world and is used to calculate the control variables in this paper.

**4.3.2 Descriptive statistics.** Table 1 reports the descriptive statistics of the main variables of the paper.

we first supplemented the correlation coefficient test, with the results shown in Table 2, revealing a positive correlation between the development of the digital economy and centralization in global value chain networks.

## 5. Empirical results and analysis

### 5.1 Baseline regression results

Table 3 reports the baseline regression results. To ensure the robustness of the results, country, industry, time, and industry-year fixed effects are included to control the impact of country-level, industry-level, and industry-changing characteristics over time in the empirical results. Column (1) considers only the effect of the single core explanatory variable of industry digital economy penetration (Digit) and it is observed that the coefficient of digital economy penetration is significantly positive at the 1% significance level, implying that every 1% increase in the level of digital economy development increases a country's value chain network centralization by 0.08%. Columns (2) and (3) are the regression results after adding a series of control variables to Column (1). Column (3) further controls industry fixed effects compared to Column (2), and it is observed that the coefficient of digital economy penetration is still significantly positive at the 1% significance level. Column (4) further controls industry-year fixed effects, and the coefficients of the core explanatory variables are still significantly positive at the 1% level, indicating that the development of the digital economy has a significant contribution to the degree of global value chain network centralization in a country. This verifies Hypothesis 1.

In addition, the regression results for the control variables are as expected. The coefficient of labor ratio (Labor) is not statistically significant, indicating that the level of a country's labor force has no significant effect on the centralization of the GVC network, and this is related to the fact that the digital economy has some substitution effect on the traditional

**Table 1. Descriptive statistics.**

| Variable | Observations | Mean | Std | Min | Max |
|---|---|---|---|---|---|
| centralization | 9625 | 6.8394 | 2.4994 | 0 | 12.8688 |
| Digit | 9625 | 0.2568 | 0.1891 | 0.0566 | 0.8807 |
| Labor | 9416 | 59.9074 | 6.2013 | 45.5233 | 74.2437 |
| Persons | 9625 | 4.5078 | 1.1999 | 1.2865 | 7.2145 |
| Scale | 9625 | 13.4145 | 3.9393 | 0 | 14.4201 |
| Fdi | 9625 | 2.5401 | 3.8481 | 0 | 14.2313 |
| GDP | 9625 | 0.0814 | 0.0041 | −0.0288 | 0.2345 |
| RTA | 9625 | 3.7367 | 0.8973 | 0.6931 | 4.4543 |


**Table 2. Correlation coefficient test.**

| | centralization | Digit | Labor | Persons | Scale | Fdi | GDP | RTA |
|---|---|---|---|---|---|---|---|---|
| centralization | 1.000 | | | | | | | |
| Digit | 0.056*** | 1.000 | | | | | | |
| Labor | 0.162*** | 0.002 | 1.000 | | | | | |
| Persons | −0.072*** | −0.001 | −0.364*** | 1.000 | | | | |
| Scale | 0.100*** | 0.000 | 0.077*** | 0.058*** | 1.000 | | | |
| Fdi | −0.025** | 0.066*** | 0.127*** | 0.000 | −0.325*** | 1.000 | | |
| GDP | 0.432*** | −0.002 | 0.343*** | 0.163*** | 0.207*** | 0.024** | 1.000 | |
| RTA | 0.099*** | 0.013 | 0.387*** | 0.183*** | 0.391*** | 0.303*** | 0.276 | 1.000 |


**Table 3. Baseline regression results of the development of the digital economy on the centralization of the value chain network.**

| | (1) | (2) | (3) | (4) |
|---|---|---|---|---|
| | FE | FE | FE | FE |
| Digit | 0.0837***(0.0294) | 0.1990***(0.0342) | 0.0857***(0.0310) | 0.0876***(0.0325) |
| Labor | | 0.0018(0.0073) | −0.0016(0.0069) | −0.0015(0.0067) |
| Persons | | −2.5502***(0.5741) | −2.1943***(0.5575) | −2.2008***(0.5537) |
| Scale | | 1.1061***(0.3982) | 1.2752***(0.3740) | 1.2734***(0.3712) |
| Fdi | | −0.0017(0.0035) | −0.0031(0.0032) | −0.0031(0.0032) |
| GDP | | 0.0814**(0.0330) | 0.1022***(0.0345) | 0.1024***(0.0348) |
| RTA | | 0.0448(0.0328) | 0.0687**(0.0321) | 0.0683**(0.0338) |
| Constant term | 6.688***(0.0682) | 15.142***(2.6211) | 13.293***(2.5762) | 13.324***(2.5692) |
| Country fixed effects | YES | YES | YES | YES |
| Industry fixed effects | YES | NO | YES | NO |
| Hausman Test | 13.54(p = 0.0003) | 14.89(p = 0.0001) | 18.99(p = 0.0000) | 12.06(p = 0.0006) |
| N | 9625 | 8818 | 8818 | 8818 |
| $R^2$ | 0.632 | 0.380 | 0.630 | 0.632 |

Note: ***, **, * represent significant at 1%, 5%, and 10% significance levels; the numbers in () represent robust standard errors; all regression results are clustered at the country-industry level; same for the latter table.


labor force. The coefficient of population density (Persons) is significantly negative, indicating that the more densely populated a country is, the less favorable it is for a country's position in the GVC network, possibly because densely populated areas tend to develop labor-intensive industries, which are not favorable as industries to improve their position in the GVC. The coefficient of industry size (Scale) is significantly positive, indicating that the larger the industry size is, the more favorable its centralization in the GVC network. The coefficient of foreign direct investment (Fdi) is not statistically significant. The coefficient of the GDP growth rate (GDP) is significantly positive, indicating that the higher the level of economic development of a country is, the more centralized it is in the GVC network, which is in line with our traditional perception. The coefficient of the number of regional trade agreements (RTA) is significantly positive, indicating that the more trade agreements a country has with other countries, the more central it is in the GVC network.

### 5.2 Treatment of endogenous problems

Due to the inherent bias in the regression model, endogeneity problems may occur in the regression results. The possible endogeneity problems are manifested in the omitted variables and reverse causality. First, regarding omitted variables, this paper not only considers the impact of the core explanatory variable of digital economy penetration on the centralization of the GVC network in the baseline regression but also adds a series of control variables and finds that the development of the digital economy promotes centralization regardless of whether the control variables are considered. This indicates that the problem of omitted variables in this paper is not serious.

Second, reverse causality is reflected in this paper as the impact of the centralization of the GVC network on the development of the digital economy. Intuitively, the reverse causality problem in this paper is also not serious because digital economy development is generally a macro strategy of a country, while the degree of value chain network centralization is often a choice of the manufacturing industry. However, for the robustness of the results, the core explanatory variable (digital economy penetration) is regressed with a one-period lag because the impact of the degree of GVC network centralization in the current period on the development of the digital economy in the previous period is almost negligible. The test results are in Columns (1) and (2) of Table 4, and the coefficients of the core explanatory variables are still significantly

positive at the 1% significance level, indicating that the results of the baseline regression are robust. It also further validates Hypothesis 1.

Either adding control variables or using lagged explanatory variables as core explanatory variables can only mitigate endogeneity to some extent. To make the findings more reliable, this paper uses instrumental variables to address potential endogeneity. The instrumental variables need to satisfy the requirements of relevance and exogeneity; referring to Nunn & Qian, the number of landline telephones in 1984 for each country is used as an instrumental variable for the development of the digital economy. First, in terms of relevance, the number of landline telephones, as an important carrier of telecommunication infrastructure construction, reflects the level of digital economy development in the past. The network can be connected by telephone dialing, so using it as an instrumental variable meets the requirement of relevance. Second, in terms of exogeneity, telephone dial-up internet access was gradually phased out after entering the new period, and 1984 is far from the sample period of this paper (2004–2014), thus meeting the requirement of exogeneity. In addition, to overcome the drawback that instrumental variables do not vary over time, this paper further refers to Nunn & Qian by regressing the interaction term between the number of landline telephones in each country in 1984 and the industry digital R&D investment in the previous year. Columns (3) and (4) of Table 4 report the results of the instrumental variables regression using two-stage least squares (2SLS), which show that the instrumental variables are significantly positive at the 1% significance level, indicating that the development of the digital economy promotes the degree of GVC network centralization, which is consistent with the baseline regression results. The Cragg-Donald Wald F statistic and Kleibergen–Paap rk Wald F statistic both exceed the 10% threshold of the Stock-Yogo test (16.38), indicating that the instrumental variables are not weak instrumental variables. The Kleibergen–Paap rk LM statistic also indicates that there is no underidentification of instrumental variables in this paper.

However, to make our research results more reliable and the causality between the development of the digital economy and centralization in global value chain networks more credible. We have selected another instrumental variable to address potential endogeneity issues. Reference to Beverelli et al [41]. we constructed a weighted value of the digital economy development level of other countries relative to the home country and the average digital economy development

**Table 4. Endogenous problem treatment.**

| | (1) | (2) | (3) | (4) |
|---|---|---|---|---|
| | **FE** | **FE** | **2SLS** | **2SLS** |
| L. Digit | 0.0822***(0.0296) | 0.0880***(0.0326) | | |
| Digit | | | 0.0937***(0.0070) | 0.0392***(0.0064) |
| Control variables | NO | YES | NO | YES |
| Country fixed effects | YES | YES | YES | YES |
| Time fixed effects | YES | YES | YES | YES |
| Industry – year fixed effects | YES | YES | YES | YES |
| Kleibergen-Paap rk LM Statistics | | | 2.5e+06***[0.000] | 2.4e+06***[0.000] |
| Cragg-Donald Wald F Statistics | | | 3.2e+06{16.38} | 2.8e+06{16.38} |
| Kleibergen-Paap rk Wald FStatistics | | | 2.5e+06{16.38} | 2.4e+06{16.38} |
| N | 8613 | 7995 | 9625 | 8818 |
| $R^2$ | 0.632 | 0.633 | | |
| **First stage regression results.** | | | | |
| Digit _IV | | | 0.9994***(0.0006) | 0.9993***(0.0006) |

Note: The numbers in () represent robust standard errors; the numbers in [] represent the p-values; the numbers in {} represent the critical value of 16.38 for the F-test of the weak instrumental variable at the 10% significance level; since the IV estimates do not report the constant term, at which point $R^2$ loses its significance, columns (3) and (4) of Table 3 do not report $R^2$.


level of countries at different income levels, multiplied by the digitalization rate of various industries in each country, serving as an instrumental variable for the digital economy at the "country-industry" level. The rationale for this "country-industry" level digital economy instrumental variable is as follows: In terms of relevance, countries with similar economic development levels tend to have similar levels of digital economy development, satisfying the relevance requirement. In terms of exogeneity, the average level of digital economy development across different income levels reflects the digital economy development within that income bracket but does not contain country-specific development characteristics. Therefore, this instrumental variable meets the requirements. Table 5 shows the regression results using the "country-industry" level instrumental variable, indicating that after addressing potential endogeneity issues, the conclusion that the development of the digital economy enhances centralization in global value chain networks still holds.

### 5.3 Robustness testing

To make the research findings more robust, this paper uses various methods to conduct robustness tests. First, the paper conducts tests to replace the explanatory variables. In the baseline regression, the explanatory variable is the GVC network centralization calculated based on the trade network and the domestic value added of exports. Compared with the trade network calculated by the traditional method, the calculation in the baseline regression better reflects the reality of today's GVC division of labor. However, to reflect the robustness of the study, this paper recalculates the GVC centralization indicator using total export trade instead of export domestic value added. The regression results are shown in Column (1) of Table 6. The coefficient of digital economy penetration is still significantly positive at the 1% significance level, indicating that the development of the digital economy continues to promote a country's central position in the GVC network.

The robustness test of replacing the core explanatory variable is manifested in the digital economy penetration rate in the baseline regression of this paper. In the robustness test, the digital economy penetration indicator (Digit 1) is recalculated using the direct coefficient depletion method instead of the full coefficient depletion method to replace the core explanatory variable in the baseline regression. Column (2) of Table 6 reports the regression results for the replacement explanatory variables, and the coefficients of the core explanatory variables are significantly positive at the 5% significance level, which is consistent with the baseline regression findings.

Considering that different trade policies have an impact on a country's position in the GVC network, this paper further controls for the impact of trade liberalization on the results of the baseline regression. The effect of trade liberalization is controlled by including the industry-level tariff levels of each country. Column (3) of Table 6 reports the impact of digital economy development on the centralization of the GVC network after controlling for trade liberalization, and the results show that the coefficient of digital economy penetration is significantly positive at the 1% significance level.

**Table 5. Regression results of instrumental variables at the "country-industry" level.**

| | (1) | (2) |
|---|---|---|
| | **2SLS** | **2SLS** |
| Digit | 0.0987***(0.0131) | 0.0881***(0.0134) |
| Control variables | NO | YES |
| Country fixed effects | YES | YES |
| Time fixed effects | YES | YES |
| Industry – year fixed effects | YES | YES |
| Kleibergen-Paap rk LM Statistics | 2.8e + 06***[0.000] | 2.7e + 06***[0.000] |
| Cragg-Donald Wald F Statistics | 3.2e + 06{16.38} | 3.4e + 06{16.38} |
| Kleibergen-Paap rk Wald FStatistics | 2.7e + 06{16.38} | 2.8e + 06{16.38} |
| N | 9625 | 8818 |


**Table 6. Robustness test results.**

| | (1) Substitution of explanatory variables | (2) Substitution of explanatory variables | (3) Consider trade liberalization | (4) Remove outliers | (5) Samples after 2008 |
|---|---|---|---|---|---|
| Digit | 0.0935***(0.0141) | | 0.0877***(0.0325) | 0.0838**(0.0326) | 0.1998***(0.0280) |
| Digit 1 | | 0.0640**(0.0278) | | | |
| Tarffic | | | 0.3241**(0.1263) | | |
| Country fixed effects | YES | YES | YES | YES | YES |
| Time fixed effects | YES | YES | YES | YES | YES |
| Industry – year fixed effects | YES | YES | YES | YES | YES |
| N | 8818 | 8818 | 8818 | 8713 | 5216 |
| $R^2$ | 0.599 | 0.632 | 0.635 | 0.632 | 0.763 |


Meanwhile, considering the impact of outliers on the regression results, this paper excludes the samples that are zero in GVC network centralization and applies a 1% tail shrinkage to the core explanatory variables and the explanatory variables. The regression results with outliers excluded are shown in Column (4) of Table 6, and the coefficient of digital economy penetration is significantly positive at the 5% significance level, further illustrating the robustness of the baseline regression results.

Finally, taking into account the impact of the 2008 financial crisis on the global economy, we re-ran the regression analysis using samples from after 2008. The results are presented in column (5) of Table 6, where the coefficient for the development of the digital economy is significantly positive at the 1% significance level, supporting the conclusions of the baseline regression.

## 6. Heterogeneity analysis and mechanism test

### 6.1 Country development and institutional quality heterogeneity

In the baseline regression, the results of this paper show that the development of the digital economy promotes the central position of a country's GVC network. To further examine the different impacts of the digital economy on countries and industries, this paper refines the study sample. First, digital economic development is closely related to a country's economic development level, and the contribution of digital economic development to the centralization of the GVC network may be different between developed and developing countries, so this paper divides the sample into developed and developing countries. Columns (1) and (2) of Table 7 report the different impacts, and the results show that digital

**Table 7. Country development and institutional quality heterogeneity.**

| | (1) Developed countries | (2) Developing countries | (3) Low institutional quality | (4) High institutional quality |
|---|---|---|---|---|
| Digit | 0.1092*(0.0572) | 0.0500(0.0433) | 0.0907**(0.0417) | 0.1814***(0.0461) |
| Control variables | YES | YES | YES | YES |
| Country fixed effects | YES | YES | YES | YES |
| Time fixed effects | YES | YES | YES | YES |
| Industry – year fixed effects | YES | YES | YES | YES |
| N | 5801 | 3017 | 5492 | 3326 |
| $R^2$ | 0.577 | 0.683 | 0.620 | 0.700 |


economic development significantly contributes to the degree of GVC network centralization in developed countries, while there is no statistically significant impact on developing countries. The reason is perhaps that developed countries, compared with developing countries, have an earlier start with respect to the digital economy, their infrastructure development is relatively well developed, and their GVC network centralization is more sensitive to the development of the digital economy.

The development of the digital economy continues to accelerate the integration of traditional economic sectors with digital technology sectors, but this effective integration relies on good institutional guarantees, so this paper considers the impact of digital economic development on GVC network centralization under different institutional qualities. The paper uses the "Solow residual" stripping method to calculate the institutional quality index for each industry in a country. The core of this method is that Solow's residual includes the technological progress factor and the institutional quality factor, as the remaining part of Solow's residual is the institutional quality factor after stripping out the technological progress factor. Industries above the mean of institutional quality are defined as high institutional quality industries, and industries below the mean are defined as low institutional quality industries. Columns (3) and (4) of Table 4 report the different effects of digital economy development on the low and high institutional quality GVC network centralization, and the results show that the coefficient of digital economy penetration is significantly positive in both low and high institutional quality samples. This indicates that digital economy development has a significant contribution to GVC network centralization under the presence of different institutional qualities. However, in terms of significance and coefficient size, digital economy development has a greater role in high institutional quality industries.

### 6.2 Heterogeneity of factor intensity

There are significant differences in the level of technology contained in different factor-intensive industries, so there are also differences in the sensitivity of the centralization of the GVC network towards the development of the digital economy in different factor-intensive industries. In this paper, the manufacturing industries involved in the WIOD are classified into three categories: labor-intensive, capital-intensive, and technology-intensive industries. The regression results are shown in Columns (1)-(3) of Table 8. The results show that the development of the digital economy contributes significantly to the GVC centralization of technology-intensive industries but it has a limited contribution in labor-intensive and capital-intensive industries. A possible reason for this is that technology-intensive industries have higher technology requirements, and the digital economy is a manifestation of this requirement, so the promotion effect is more obvious in technology-intensive industries. In labor-intensive industries, digital economic development may have a certain substitution effect on traditional labor. In capital-intensive industries, digital economic development may have a "crowding-out effect" on capital investment, and thus, in these two types of industries, digital economic development cannot significantly contribute to the centralization of the GVC network.

**Table 8. Heterogeneity of different factor-intensive industries.**

| | (1) | (2) | (3) |
|---|---|---|---|
| | **Labor-intensive** | **Capital-intensive** | **Technology-intensive** |
| Digit | 0.0199(0.0613) | 0.0344(0.0479) | 0.161$^{***}$(0.0349) |
| Control variables | YES | YES | YES |
| Country fixed effects | YES | YES | YES |
| Time fixed effects | YES | YES | YES |
| Industry – year fixed effects | YES | YES | YES |
| N | 1140 | 2660 | 2660 |
| $R^2$ | 0.736 | 0.655 | 0.771 |


### 6.3 Mechanism test

The previous study has illustrated that the development of a country's digital economy promotes its centralization in the GVC network, and based on this, the sample is divided to analyze the different impacts of digital economic development on the centralization of the GVC network in different countries with different levels of economic development, institutional quality, and factor-intensity industries. At the same time, the channels through which digital economy development affects centralization are of interest to this paper. Based on previous theoretical studies, this paper argues that digital economic development affects the centralization of the GVC network in each country through the productivity channel as well as the resource allocation channel.

Regarding the choice of the mechanism test method, since the mediation model was proposed, it has been rapidly and commonly applied in economics research, but there are certain problems in this regard. In terms of application scenarios, the mediation model is more suitable for psychological studies; in terms of technical aspects, the mediation model has the problem of endogeneity and omission of variables. Therefore, this paper refers to Huang & Song and constructs the following mechanism test model to study the mechanism of the influence of digital economy development on the centralization of the GVC network.

$$mech_{cit} = \alpha_0 + \alpha_1 digit_{cit} + \alpha \sum control_{cit} + \gamma_c + \delta_i + \theta_t + \varepsilon_{cit} \quad (22)$$

where $mech_{cit}$ represents the two mechanism variables (productivity and resource allocation), and the rest of the variables are defined as above. The productivity (Tfp) of each industry is calculated by the "Solow residual method", which is obtained from the WIOD database; based on the calculation of resource allocation (DisIntfp), this paper refers to Hsieh & Klenow and Jiang et al., viewing resource mismatch as a deviation of productivity from the mean [42]. Therefore, this paper uses the variance of the log of industry productivity (a larger value indicates a higher degree of industry resource mismatch) to measure the degree of resource mismatch in a country based on the calculated productivity of each industry in a country.

Table 9 reports the results of the mechanism test, Columns (1) and (2) report the impact of digital economy development on productivity, and the coefficients of digital economy penetration are both significantly positive at the 1% level of significance, indicating that digital economy development significantly enhances the productivity of a country's industry. It is clear that digital economic development achieves its facilitating effect on the centralization of the GVC network by enhancing the productivity of industries within a country. Columns (3) and (4) report the effect of digital economic development on resource mismatch, and the coefficients of digital economic penetration are both significantly negative at the 1% significance level, indicating that digital economic development facilitates the mitigation of resource mismatch

**Table 9. Mechanism test.**

| | (1) | (2) | (3) | (4) |
|---|---|---|---|---|
| | **Productivity Channel** | | **Resource allocation channel** | |
| | **Tfp** | | **DisIntfp** | |
| Digit | 0.0060***(0.0018) | 0.0055***(0.0019) | −0.0022***(0.0003) | −0.0017***(0.0003) |
| Control variables | NO | YES | NO | YES |
| Country fixed effects | YES | YES | YES | YES |
| Industry fixed effects | YES | YES | YES | YES |
| Time fixed effects | YES | YES | YES | YES |
| N | 9625 | 8818 | 9625 | 8818 |
| $R^2$ | 0.422 | 0.431 | 0.945 | 0.949 |


in industries. The alleviation of resource mismatch is conducive to improving the efficiency of resource allocation at the national level, which in turn is more conducive to becoming more competitive. Thus, digital economy development promotes the centralization of a country in the GVC network by reducing the degree of resource mismatch. The above findings validate Hypotheses 2 and 3 in the theoretical part of the paper.

## 7. Further analysis: The impact of digital economy development on the length and location of the GVC

This paper mainly studies the impact of digital economy development on the centralization of a country's GVC network. To more deeply analyze the impact of digital economic development on countries' competition in GVC, this paper further decomposes GVC specifically into length and location.

The first is the impact of digital economy development on the length of GVC; this paper refers to Wang et al., using the sum of the average forward and average backward GVC lengths as the industry-level GVC length (GVC_ length) of a country [43], where the forward average production length consists of four components: forward pure domestic value chain length, forward traditional trade value chain length, forward simple GVC length, and forward complex GVC length; the backward average production length consists of four components: backward pure domestic value chain length, backward traditional trade value chain length, backward simple GVC length, and backward complex GVC length. The data for calculating the industry-level GVC lengths are obtained from the UIBE GVC Indicators database. Columns (1) and (2) of Table 10 report the impact of digital economy development on value chain length, and the study finds that the coefficient of the core explanatory variable of digital economy penetration is significantly positive at the 1% significance level regardless of whether the control variables are considered, indicating that digital economy development lengthens the value chain length of each industry. Thus, digital economy development not only promotes the degree of GVC network centralization but it also enhances the production chains of countries and extends their length in the GVC.

This is followed by the impact of digital economy development on the location of GVC, which is calculated using the value added method (GVC_ location), with the following formula:

$$GVC_location = ln\left(1 + \frac{DVA_INT + DVA_REX}{E}\right) - ln\left(1 + \frac{FVA_FIN + FVA_INT}{E}\right) \tag{23}$$

where $GVC_location$ represents GVC location; $E$ represents industry export trade volume; $DVA_INT$ represents domestic value added exported as direct intermediate goods; $DVA_REX$ represents domestic value added exported as indirect intermediate goods; $FVA_FIN$ represents foreign value added exported as final goods; and $FVA_INT$ represents foreign value added exported as intermediate goods. Columns (3) and (4) of Table 10 report the impact of digital economy development on the location of the GVC, and the coefficient of digital economy penetration is significantly positive at the 1%

**Table 10. Impact of digital economy development on the length and location of the value chain.**

| | (1) | (2) | (3) | (4) |
|---|---|---|---|---|
| | **Value chain length** | | **Value Chain Location** | |
| | **GVC_ length** | | **GVC_ location** | |
| Digit | 0.0093***(0.0029) | 0.0092***(0.0030) | 0.1030***(0.0211) | 0.1031***(0.0212) |
| Control variables | NO | YES | NO | YES |
| Country fixed effects | YES | YES | YES | YES |
| Time fixed effects | YES | YES | YES | YES |
| Industry – year fixed effects | YES | YES | YES | YES |
| N | 9424 | 8626 | 9625 | 8818 |
| $R^2$ | 0.752 | 0.758 | 0.594 | 0.595 |


level of significance regardless of whether the control variables are considered. In addition to promoting the centralization of the GVC network, digital economic development drives a country's industries to move up in the GVC.

## 8. Conclusions and policy recommendations

Based on a heterogeneous firm export model and with reference to Bai et al. in terms of introducing productivity and resource mismatch, this paper investigates the impact of the digital economy on firm export behavior and aggregates the impact at the national level, thus constructing a theoretical framework for the impact of digital economy development on GVC network centralization. Furthermore, the UIBE GVC Indicators database and the WIOD database are used to calculate the digital economy penetration rate and the GVC network centralization indicators, and the impact of digital economy development in each country and the endogenous mechanism are empirically examined. In contrast to existing research, which focuses more on the impact of the digital economy on the upgrading of GVCs and ignores the impact on the centrality of GVC network, the findings of this paper are an important addition in this regard. The study concludes that, first, digital economic development is conducive to enhancing a country's central position in the GVC network, and the conclusion still holds after considering the treatment of endogeneity issues and a series of robustness tests. Second, heterogeneity analysis shows that digital economic development contributes more to the centralization of the GVC network in developed countries than in developing countries; digital economic development contributes more in high institutional quality industries than in low institutional quality industries; and digital economic development contributes more in technology-intensive industries than in capital-intensive and labor-intensive industries. Third, the mechanism analysis shows that digital economic development achieves a country's centralization in the GVC network by enhancing productivity and alleviating labor resource mismatch. Fourth, the expansion analysis shows that digital economy development not only promotes centralization but it also extends the length of the GVC in each industry and enhances the upstream part of the GVC.

The findings of this paper have important policy implications. First, the digital economy is actively developing. The research in this paper finds that with the development of the digital economy, a country's position in the global value chain network continues to be centralized and moves up the value chain. Therefore, countries should continue to launch policies to support the development of the digital economy, improve the construction of digital infrastructure, and help enterprises realize digital transformation. Second, the digital economy should be relied on to transform the mode of development. The traditional mode of economic growth driven by capital factors, natural resources and labor inputs is gradually being replaced by the mode of growth driven by science and technology and innovation. Frontier science and technology, represented by the digital economy, has become increasingly important in economic development, such as the use of digital finance to continuously optimize the efficiency of resource allocation [44]. By promoting the rapid development of the digital economy, not only can it effectively reduce labor costs, but it can also alleviate a country's resource mismatch and build strong support for countries to participate in the competitive heights of the global value chain network. Finally, in response to the current situation of developing countries in the development of the digital economy and to enhance their position in the global value chain network, this paper recommends that emphasis should be placed on strengthening digital infrastructure construction. Measures such as expanding high-speed internet coverage and enhancing data center capabilities should be taken to provide solid support for the development of the digital economy. Simultaneously, it is crucial to optimize the intellectual property protection system, establish a legal and regulatory framework in line with international standards, strengthen law enforcement, and actively promote the deep integration of digital technologies with the manufacturing sector. This will stimulate innovation vitality and attract foreign investment. Through policy guidance and financial support, traditional manufacturing industries can be assisted in transforming and upgrading, thereby enabling developing countries to occupy a more advantageous position in the global value chain.

There are various limitations as well as the space that can be improved in the future. First, the object of this paper's research is GVC network centralization at the country-industry level, whereas in today's global world, enterprises are the main body participating in the competition of GVC, and the development of the digital economy effects on enterprises is

the direction we need to study. Second, due to the availability of data, the research interval of this paper covers 2004–2014. However, with the continuous development of the digital economy, its long-term impact on the centralization of global value networks has to be further refined and analyzed after the database is updated.

## Supporting information

**S1. Appendix.**
(DOCX)

## Author contributions

**Conceptualization:** Hankun Yuan.

**Data curation:** Mingyang Yue.

**Formal analysis:** Hankun Yuan.

**Investigation:** Mingyang Yue.

**Methodology:** Hankun Yuan.

**Project administration:** Mingyang Yue.

**Resources:** Mingyang Yue.

**Software:** Rong Wang.

**Supervision:** Rong Wang.

**Validation:** Biao Ren.

**Visualization:** Xueyang Li.

**Writing – original draft:** Hankun Yuan.

**Writing – review & editing:** Biao Ren.

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
