## [Decision Letter · Decision Letter 0]

PONE-D-23-10130Digital Economy Development and Global Value Chain Network CentralizationPLOS ONE

Dear Dr. yue,

Thank you for submitting your manuscript to PLOS ONE. After careful consideration, we feel that it has merit but does not fully meet PLOS ONE’s publication criteria as it currently stands. Therefore, we invite you to submit a revised version of the manuscript that addresses the points raised during the review process.

The research on digital economy development and its significance in enhancing a country's central location in the GVC network is important. The findings of the study are interesting and contribute to the literature. The research is sound and has possible policy implications, but it needs major revisions, which are presented at the end of this email.

We look forward to receiving your revised manuscript.

Kind regards,

Imran Ur Rahman, Ph.D

Academic Editor

PLOS ONE

Additional Editor Comments:

1. The introduction section should further clarify several questions: Why is the topic significant in the current economic landscape? What are the key research questions and problems? I would suggest the author enhance the theoretical argument. This will help readers who are not familiar with the topic to better understand the research objectives.

2. The literature review should be enhanced by adding a separate section or more literature studies and providing relevant citations. It can be further enriched with up-to-date studies to motivate the readers in the background and currency of the subject of the research. Moreover, the authors should further clarify the gaps in the literature and how this research contributes to filling those gaps.

3. The methodology section is good; however, the authors should further clarify and justify their choice of methods used. In addition, the author should also explain the choice of the Fixed Effect (FE) model.

4. Presentation of the summary of descriptive statistics in a tabulated form is also recommended, which is missing in the article.

5. The author should provide references and links for all the sources used in the research. For instance, the references for the UIBE database and WIOD database are missing.

6. The language of the paper needs careful editing and proofreading to improve readability.

7. Please enhance the conclusion by further clarifying the limitations and future research directions.

Reviewers' comments:

Reviewer's Responses to Questions

**Comments to the Author**

1. Is the manuscript technically sound, and do the data support the conclusions?

Reviewer #1: Yes

Reviewer #2: Yes

2. Has the statistical analysis been performed appropriately and rigorously? 

Reviewer #1: No

Reviewer #2: Yes

3. Have the authors made all data underlying the findings in their manuscript fully available?

Reviewer #1: No

Reviewer #2: Yes

4. Is the manuscript presented in an intelligible fashion and written in standard English?

Reviewer #1: No

Reviewer #2: Yes

5. Review Comments to the Author

Reviewer #1: Title: Digital Economy Development and Global Value Chain Network Centralization

The development of digital economy is an important reliance for promoting high-quality opening up and achieving high-end embedding of manufacturing into global value chain�GVC�. Based on the heterogeneous enterprise model, this paper introduces the factors of digital economy development, aggregates the production behavior of enterprises to the national level, and empirically examines the impact of digital economy development on the centralization of GVC network and the internal mechanism. The result shows that digital economy development significantly enhances a country's central location in the GVC network, and the contribution is mainly realized through productivity enhancement and mitigation of resource mismatch. Compared with developing countries and low institutional quality industries, digital economy development contributes more to the centralization of GVC network in developed countries and high institutional quality industries. Further analysis shows that digital economy development not only enhances a country's central location in the GVC network, but also extends the length of a country's GVC as well as promotes a country to climb up the GVC.

1. In order to establish the significance and value of the study, it is necessary to provide a comprehensive rationale for the research, which emphasizes its relevance and unique contributions to the current scholarly discourse. This will strengthen the study's originality and scholarly impact. Following are useful suggested studies to get the benefit to update this part:

https://doi.org/10.1007/s11356022197186 ;https://doi.org/10.3390/su14031054;https://doi.org/10.1007/s1135602117438x;https://doi.org/10.1016/j.resourpol.2022.102730 ;https://doi.org/10.1108/FS-02-2021-0053; https://doi.org/10.1002/ijfe.2073; https://doi.org/10.1108/LHT-03-20210113;https://doi.org/10.1007/s1135602221929w;https://doi.org/10.3389/fpubh.2022.1009393;https://doi.org/10.1002/sd.2529;https://doi.org/10.3389/fenvs.2022.1068398;https://doi.org/10.1016/j.techfore.2023.122413; https://doi.org/10.1007/s11356-022-20178-1

2. The authors are requested to provide a thorough justification for the choice of the current methodology, including an explanation of the benefits and advantages it offers over alternative approaches. This will lend greater credibility and rigor to the study's research design.

3. To highlight the novelty and contribution of the current study, it is necessary to outline the ways in which it diverges from the existing body of literature. Therefore, the concluding section of the literature review should include a clear differentiation of the study from prior research.

4. I would kindly request that the authors re-examine the policy implications derived from the study's findings, to ensure their accuracy and relevance to the study's objectives.

6. In order to ensure the relevance and timeliness of the study’s cited sources, it is recommended that the authors review and update the references section.

7. The study appears to employ an excessive amount of acronyms, which may impede comprehension for the wider readership of the journal. Thus, it is recommended that the authors avoid excessive usage of acronyms and instead use the full form of terms to ensure ease of understanding for readers."

Reviewer #2: The paper includes the very detailed description of the applied quantitative model, however there are a few references in the text to underlying models (such as MO model, CES form) which are not explained for the readers, however at least a short footnote would be desirable. The quantitative analysis included 3 hypotheses, but the discussion of the results or the conclusions fails to come back to these hypotheses and discuss them in a "concentrated" way. In my opinion this would also strengthen the soundness of the results. The last part of the conclusions includes a couple of policy recommendations which are only very loosely related to the quantitative analysis presented in the paper. Please review/reconsider them and make them more related to the empirical evidences.

6. PLOS authors have the option to publish the peer review history of their article (what does this mean? ). If published, this will include your full peer review and any attached files.

**Do you want your identity to be public for this peer review?** For information about this choice, including consent withdrawal, please see our Privacy Policy .

Reviewer #1: **Yes: ** KASHIF ABBASS

Reviewer #2: No

<quillbot-extension-portal></quillbot-extension-portal>

---

## [Author Response · Author response to Decision Letter 1]

17 Aug 2023

Response to Editor and Reviewers

Dear Editor and Reviewers:

First of all, please allow us to take this opportunity to express our heartfelt thanks to you for taking time out from your busy schedule to review this manuscript. You have provided us with constructive comments and suggestions, which are of great help for us to further improve this manuscript. We have carefully reviewed and revised the manuscript according to your valuable comments and suggestions. Here, we explain the revised work in detail below and provide the point-by-point responses to the editor’s and reviewers’ comments. Moreover, we have included continuous line numbers in the revised manuscript and have marked all variations in blue.

Response to Editor:

(1) The introduction section should further clarify several questions: Why is the topic significant in the current economic landscape? What are the key research questions and problems? I would suggest the author enhance the theoretical argument. This will help readers who are not familiar with the topic to better understand the research objectives.

Response: Thank you for your specific and professional comments and suggestions. We have revised the introduction to emphasize the research significance and key questions of the article, as shown in blue below.

It is observed that enhancing the competitiveness of countries in global value chains requires relying on the support of high technology represented by the digital economy, this provides a rich practical significance and realistic background for the research of this paper.

Whereas the realization of GVC network centralization is a manifestation of countries' competitiveness in the GVC system; unfortunately, most studies currently focus on the drive of the digital economy for upgrading the GVC and have not yet deeply examined the impact of the digital economy on the centralization of the GVC network. Then, the key questions of this research are as follows: with the development of the global digital economy, what is its impact on the position of countries in the global value chain network? What kind of mechanism of action drives this impact? The answers to these questions are conducive to promoting the high-quality development of the manufacturing industry and realizing a higher level of integration into the global value chain network.

(2) The literature review should be enhanced by adding a separate section or more literature studies and providing relevant citations. It can be further enriched with up-to-date studies to motivate the readers in the background and currency of the subject of the research. Moreover, the authors should further clarify the gaps in the literature and how this research contributes to filling those gaps.

Response: Thank you for your specific and professional comments and suggestions. We have included the literature review as a separate section and added the corresponding literature. Our contribution to the existing research has also been added at the end of the literature review. The modifications are shown in blue below.

Ⅱ. Literature Review

The research related to this paper focuses on the following aspects: first, the changes in the division of production labor in today's GVC. The change in firms in terms of the division of production in GVC has been measured using the concept of the number of production stages[15]. Currently, the division of labor in GVC is facing many challenges, and COVID-19 has had a great impact on the global production supply chain [1,12,28], as it slowed the growth of the global economy and gradually reduced the incentive of countries to participate in the division of labor in GVC. The rise of artificial intelligence (AI) has had an impact on GVC, as the application of AI increased the efficiency of using intermediate goods to produce final goods [10], making the production process more automated and smarter, saving labor costs [5], and improving the division of labor of firms in GVC.

In contrast to the current literature, this paper investigates the impact of digital economy development on the centralization of GVC networks from theoretical and empirical aspects and makes innovations in the following three aspects.

First, this paper extends the heterogeneous firm trade model, introduces digital economy development factors in firms' production behavior and export decisions, and gauges firm behavior at the country level to construct a theoretical framework of the impact of digital economy development on the centralization of GVC networks.

Second, this paper uses country-industry level data to empirically study how digital economy development affects the centralization of the GVC network. Referring to Arnold et al. and Xiao et al., this paper uses the UIBE GVC Indicators database and the WIOD database to construct country-industry level digital economy development and GVC network centralization indicators [9,37]. Considering possible endogeneity problems, this paper refers to Nunn & Qian and uses the number of landline telephones in each country in 1984 as an instrumental variable for the least squares analysis [29].

Third, this paper summarizes two major channels through which the development of the digital economy affects the centralization of the GVC network, namely, the productivity channel and the resource allocation channel. Numerous studies have shown that digital economic development effectively reduces export fixed costs, while the digital economy has a significant substitution effect on labor and it increases labor productivity. With the increase in productivity, a country has a more competitive advantage in the GVC network. In addition, the development of the digital economy facilitates the free flow of production factors among national sectors, alleviating resource mismatch, enabling advantage gains for a country in global competition, and promoting such a country to the center of the GVC network.

(3) The methodology section is good; however, the authors should further clarify and justify their choice of methods used. In addition, the author should also explain the choice of the Fixed Effect (FE) model.

Response: Thank you for your specific and professional comments and suggestions. We state the reason for the choice of method, with modifications as shown in blue below.

(i) Method selection and model setting

Making a comprehensive assessment, this paper chooses the panel fixed-effects model (FE) to test the impact of digital economy development on the centralization of global value chain networks. There are three main reasons for choosing this method. First, from the data point of view, this paper mainly uses cross-country panel data from 2004-2014 for the study, which is in line with the conditions for the use of panel models. Second, compared with ordinary least squares (OLS) and the panel random effects model, the panel fixed effects model considers individual and time effects, making the parameter estimation more accurate. Third, this paper further uses the Hausman test to determine that it is more reasonable to use the panel fixed effects model. Therefore, this paper uses the panel fixed effects model as the main method of empirical research.

(4) Presentation of the summary of descriptive statistics in a tabulated form is also recommended, which is missing in the article.

Response: Thank you for your specific and professional comments and suggestions. We have added a descriptive statistics section to the revised draft. The modifications are shown in blue below.

(2) Descriptive statistics

Table 1 reports the descriptive statistics of the main variables of the paper.

Table 1. Descriptive statistics

Variable Observations Mean Std Min Max

centralization 9625 6.8394 2.4994 0 12.8688

Digit 9625 0.2568 0.1891 0.0566 0.8807

Labor 9416 59.9074 6.2013 45.5233 74.2437

Persons 9625 4.5078 1.1999 1.2865 7.2145

Scale 9625 13.4145 3.9393 0 14.4201

Fdi 9625 2.5401 3.8481 0 14.2313

GDP 9625 0.0814 0.0041 -0.0288 0.2345

RTA 9625 3.7367 0.8973 0.6931 4.4543

(5) The author should provide references and links for all the sources used in the research. For instance, the references for the UIBE database and WIOD database are missing.

Response: Thank you for your specific and professional comments and suggestions. Forgive us for not paying attention to this literature, which we have added to the revised manuscript and revised elsewhere accordingly. In addition, we also add more literature as support, as follows.

1. Timmer M P, Dietzenbacher E, Los B, Stehrer R, De Vries G J. An Illustrated User Guide to the World Input–Output Database: the Case of Global Automotive Production[J]. Review of International Economics, 2015, 23: 575–605.

2. Xiao H Sun T Meng B Cheng L. Complex Network Analysis for Characterizing Global Value Chains in Equipment Manufacturing[J]. Plos One, 12 (1), e169549.

(6) The language of the paper needs careful editing and proofreading to improve readability.

Response: Thank you for your specific and professional comments and suggestions. We carefully proofread and optimize the language of writing. Here are the proofs

(7) Please enhance the conclusion by further clarifying the limitations and future research directions.

Response: Thank you for your specific and professional comments and suggestions. We have added the limitations of the article and room for research directions in the conclusion section. The modifications are shown in blue below.

There are various limitations as well as the space that can be improved in the future. First, the object of this paper's research is GVC network centralization at the country-industry level, whereas in today's global world, enterprises are the main body participating in the competition of GVC, and the development of the digital economy effects on enterprises is the direction we need to study. Second, due to the availability of data, the research interval of this paper covers 2004-2014. However, with the continuous development of the digital economy, its long-term impact on the centralization of global value networks has to be further refined and analyzed after the database is updated.

(8) In your Data Availability statement, you have not specified where the minimal data set underlying the results described in your manuscript can be found.

Response: Thank you for your specific and professional comments and suggestions. We added the Data Availability statement:

Data Availability Statement

WIOD database: https://www.rug.nl/ggdc/valuechain/wiod/

UIBE GVC Indicators database: http://gvcdb.uibe.edu.cn/

all data are fully available without restriction

Response to Reviewer #1:

(1) In order to establish the significance and value of the study, it is necessary to provide a comprehensive rationale for the research, which emphasizes its relevance and unique contributions to the current scholarly discourse. This will strengthen the study's originality and scholarly impact. Following are useful suggested studies to get the benefit to update this part:

https://doi.org/10.1007/s11356022197186 ;

https://doi.org/10.3390/su14031054;

https://doi.org/10.1007/s1135602117438x;

https://doi.org/10.1016/j.resourpol.2022.102730 ;

https://doi.org/10.1108/FS-02-2021-0053;

https://doi.org/10.1002/ijfe.2073;

https://doi.org/10.1108/LHT-03-20210113;

https://doi.org/10.1007/s1135602221929w;

https://doi.org/10.3389/fpubh.2022.1009393;

https://doi.org/10.1002/sd.2529;

https://doi.org/10.3389/fenvs.2022.1068398;

https://doi.org/10.1016/j.techfore.2023.122413;

https://doi.org/10.1007/s11356-022-20178-1.

Response: Thank you for your specific and professional comments and suggestions. Forgive us for not paying attention to this literature, which we have added to the revised manuscript and revised elsewhere accordingly. In addition, we also add more literature as support, as follows:

1. Abbass K, Begum H, Alam ASAF, Awang AH, Abdelsalam MK, Egdair IMM, Wahid R. Fresh Insight through a Keynesian Theory Approach to Investigate the Economic Impact of the COVID-19 Pandemic in Pakistan[J]. Sustainability, 2022, 14(3):1054.

2. Acemoglu D, Restrepo P. The Race between Man and Machine Implications of Technology for Growth Factor Shares�and Employment�J�. American Economic Review�2018�108�6��1488-1542.

3. Aghion P, Howitt P. The Economics of Growth�M�. Massuchusetts The MIT Press�2008.

4. Begum H , Abbass K , Alam A S A F , Song H, Chowdhury MT Ghani ABA. Impact of the COVID-19 pandemic on the environment and socioeconomic viability: a sustainable production chain alternative[J]. Foresight:The journal for future studies, strategic thinking and policy, 2022(3/4):24.

5. Fally�T. Production Staging Measurement and Facts�R�. University of Colorado Boulder Working Paper�2012.

6. Naseer S, Khalid S, Parveen S, Abbass K, Song Hand Achim MV (2023) COVID-19 outbreak:Impact on global economy[J]. Frontiers in Public Health 2022, (10):1-13.

(2) The authors are requested to provide a thorough justification for the choice of the current methodology, including an explanation of the benefits and advantages it offers over alternative approaches. This will lend greater credibility and rigor to the study's research design.

Response: Thank you for your specific and professional comments and suggestions. We state the reason for the choice of method, with modifications as shown in blue below.

(i) Method selection and model setting

Making a comprehensive assessment, this paper chooses the panel fixed-effects model (FE) to test the impact of digital economy development on the centralization of global value chain networks. There are three main reasons for choosing this method. First, from the data point of view, this paper mainly uses cross-country panel data from 2004-2014 for the study, which is in line with the conditions for the use of panel models. Second, compared with ordinary least squares (OLS) and the panel random effects model, the panel fixed effects model considers individual and time effects, making the parameter estimation more accurate. Third, this paper further uses the Hausman test to determine that it is more reasonable to use the panel fixed effects model. Therefore, this paper uses the panel fixed effects model as the main method of empirical research.

(3) To highlight the novelty and contribution of the current study, it is necessary to outline the ways in which it diverges from the existing body of literature. Therefore, the concluding section of the literature review should include a clear differentiation of the study from prior research.

Response: Thank you for your specific and professional comments and suggestions. We have included the literature review as a separate section and added the corresponding literature. Our contribution to the existing research has also been added at the end of the literature review. The modifications are shown in blue below.

Ⅱ. Literature Review

The research related to this paper focuses on the following aspects: first, the changes in the division of production labor in today's GVC. The change in firms in terms of the division of production in GVC has been measured using the concept of the number of production stages[15]. Currently, the division of labor in GVC is facing many challenges, and COVID-19 has had a great impact on the global production supply chain [1,12,28], as it slowed the growth of the global economy and gradually reduced the incentive of countries to participate in the division of labor in GVC. The rise of artificial intelligence (AI) has had an impact on GVC, as the application of AI increased the efficiency of using intermediate goods to produce final goods [10], making the production process more automated and smarter, saving labor costs [5], and improving the division of labor of firms in GVC.

In contrast to the current literature, this paper investigates the impact of digital economy development on the centralization of GVC networks from theoretical and empirical aspects and makes innovations in the following three aspects.

First, this paper extends the heterogeneous firm trade model, introduces digital economy development factors in firms' production behavior and export decisions, and gauges firm behavior at the country level to construct a theoretical framework of t

---

## [Decision Letter · Decision Letter 1]

PONE-D-23-10130R1Digital Economy Development and Global Value Chain Network CentralizationPLOS ONE

Dear Dr. yue,

Thank you for submitting your manuscript to PLOS ONE. After careful consideration, we feel that it has merit but does not fully meet PLOS ONE’s publication criteria as it currently stands. Therefore, we invite you to submit a revised version of the manuscript that addresses the points raised during the review process.

For detailed comments of the editors and reviewers, please refer to 'Additional Editor Comments' provided at the end of this email.

We look forward to receiving your revised manuscript.

Kind regards,

Imran Ur Rahman, Ph.D

Academic Editor

PLOS ONE

Journal Requirements:

Additional Editor Comments:

I appreciate the efforts of the authors in the revision of the manuscript. Although most of the comments have been addressed, there are still some minor points that need to be revised.

1. The introduction section is clear, but it can be further developed. I would suggest the author to further enhance the theoretical argument.

2. Please enhance the literature further and provide more relevant references for some contemporary literature works.

3. The authors should revise the language and formatting of the article. Please follow the guidelines of PLOS ONE for all the headings and sub-headings as well as uniformity of the text and tables.

Reviewers' comments:

Reviewer's Responses to Questions

**Comments to the Author**

1. If the authors have adequately addressed your comments raised in a previous round of review and you feel that this manuscript is now acceptable for publication, you may indicate that here to bypass the “Comments to the Author” section, enter your conflict of interest statement in the “Confidential to Editor” section, and submit your "Accept" recommendation.

Reviewer #2: All comments have been addressed

Reviewer #3: (No Response)

Reviewer #4: (No Response)

2. Is the manuscript technically sound, and do the data support the conclusions?

Reviewer #2: Yes

Reviewer #3: Yes

Reviewer #4: Yes

3. Has the statistical analysis been performed appropriately and rigorously? 

Reviewer #2: Yes

Reviewer #3: Yes

Reviewer #4: Yes

4. Have the authors made all data underlying the findings in their manuscript fully available?

Reviewer #2: Yes

Reviewer #3: Yes

Reviewer #4: No

5. Is the manuscript presented in an intelligible fashion and written in standard English?

Reviewer #2: Yes

Reviewer #3: Yes

Reviewer #4: Yes

6. Review Comments to the Author

Reviewer #2: The authors have addressed all my comments from the previous round of review and this second version of the paper has been enhanced a lot.

Reviewer #3: It can be seen that the author spent a lot of time revising this manuscript this time, and overall, the manuscript basically meets the publishing requirements. However, there are still the following issues:

1. The introduction section needs to be strengthened, and the most important thing is to highlight the research background, theoretical framework, and existing research deficiencies in the introduction section

2. The conclusion section of the study still needs to be strengthened. In addition to summarizing this article, the conclusion of the study also needs to be discussed in relation to existing relevant research. And the differences between your research findings and existing related research.

3. I think the policy recommendations in this manuscript are not sufficient, It is recommended to refer to or cite this paper: https://doi.org/10.1016/j.resourpol.2023.104174

Reviewer #4: (1) It is suggested to further emphasize the significance of GVC network centralization in the introduction.

(2) It is suggested to focus on the impact of ICT, network infrastructure, informatization and other economic variables on GVC in the literature review, the existing references pay less attention to the core issues of this paper

7. PLOS authors have the option to publish the peer review history of their article (what does this mean? ). If published, this will include your full peer review and any attached files.

**Do you want your identity to be public for this peer review?** For information about this choice, including consent withdrawal, please see our Privacy Policy .

Reviewer #2: No

Reviewer #3: No

Reviewer #4: No

---

## [Author Response · Author response to Decision Letter 2]

1 May 2024

Response to Editor and Reviewers

Dear Editor and Reviewers:

First of all, please allow us to take this opportunity to express our heartfelt thanks to you for taking time out from your busy schedule to review this manuscript. You have provided us with constructive comments and suggestions, which are of great help for us to further improve this manuscript. We have carefully reviewed and revised the manuscript according to your valuable comments and suggestions. Here, we explain the revised work in detail below and provide the point-by-point responses to the editor’s and reviewers’ comments. Moreover, we have included continuous line numbers in the revised manuscript and have marked all variations in red.

Response to Editor:

(1) The introduction section is clear, but it can be further developed. I would suggest the author to further enhance the theoretical argument.

Response: Thank you for your specific and professional comments and suggestions. We have added to the theoretical part of the introduction to strengthen the theoretical discussion:

In terms of theoretical research, this paper proves that the development of the digital economy promotes the centrality of the global value chain network by constructing a heterogeneous enterprise model, and that this promotion is mainly realized through the improvement of productivity and the reduction of resource allocation efficiency. With the development of the digital economy, it creates advantages for enterprises to participate in global value chain competition, and therefore promotes the country's production and cooperation in the global value chain network, and promotes its gradual development towards the center of the global value chain network.

(2) Please enhance the literature further and provide more relevant references for some contemporary literature works.

Response: Thank you for your specific and professional comments and suggestions. We have added the references accordingly in the References section, and the specific changes are described in the References section.

(3)The authors should revise the language and formatting of the article. Please follow the guidelines of PLOS ONE for all the headings and sub-headings as well as uniformity of the text and tables.

Response: Thank you for your specific and professional comments and suggestions. We have prepared all headings and sub-headings in accordance with PLOS ONE guidelines and harmonized the text and tables. The changes are shown in the revised version.

Response to Reviewer :

(1)The introduction section needs to be strengthened, and the most important thing is to highlight the research background, theoretical framework, and existing research deficiencies in the introduction section.

Response: Thank you for your specific and professional comments and suggestions. We make additions to the introduction section, mainly adding the theoretical framework and existing research deficiencies. The specific modifications are as follows:

In terms of theoretical research, this paper proves that the development of the digital economy promotes the centrality of the global value chain network by constructing a heterogeneous enterprise model, and that this promotion is mainly realized through the improvement of productivity and the reduction of resource allocation efficiency. With the development of the digital economy, it creates advantages for enterprises to participate in global value chain competition, and therefore promotes the country's production and cooperation in the global value chain network, and promotes its gradual development towards the center of the global value chain network.

This paper explores the impact of the development of the digital economy on the centralization of GVC networks mainly through theory and empirical evidence, and we find that, first, the digital economy significantly enhances the status of countries in GVC networks. Second, this enhancing effect varies with the degree of economic development and institutional environment of each country. Third, the development of the digital economy achieves the centralization of GVC networks through productivity enhancement and resource mismatch mitigation. Fourth, the digital economy effectively enhances a country's breadth and depth in global value chains. In addition, there is room to expand the research in this paper, such as considering the impact of GVC network centrality from the firm level.

(2) The conclusion section of the study still needs to be strengthened. In addition to summarizing this article, the conclusion of the study also needs to be discussed in relation to existing relevant research. And the differences between your research findings and existing related research.

Response: Thank you for your specific and professional comments and suggestions. We have modified the findings of the study to emphasize the linkage of the findings to existing research, as described below:

In contrast to existing research, which focuses more on the impact of the digital economy on the upgrading of GVCs and ignores the impact on the centrality of GVC networks, the findings of this paper are an important addition in this regard.

(3) I think the policy recommendations in this manuscript are not sufficient, It is recommended to refer to or cite this paper: https://doi.org/10.1016/j.resourpol.2023.104174

Response: Thank you for your specific and professional comments and suggestions. We have cited the above literature in our policy recommendations. The specific changes are set out below:

The traditional mode of economic growth driven by capital factors, natural resources and labor inputs is gradually being replaced by the mode of growth driven by science and technology and innovation. Frontier science and technology, represented by the digital economy, has become increasingly important in economic development, such as the use of digital finance to continuously optimize the efficiency of resource allocation[38].

38. Huang A Z., Bi Q X., Dai L T., et al. Investigating the impact of financial development on the resource curse form its dual effect[J]. Resources Policy, 2023,86(10):1403-1448.

(4) It is suggested to further emphasize the significance of GVC network centralization in the introduction.

Response: Thank you for your specific and professional comments and suggestions. We have emphasized the significance of GVC networks centralization in the introduction. Specific concrete changes are described below:

The global value chain network not only reflects the overall world trade pattern and trade connections but also demonstrates the importance of countries in global trade and whether they are in pivotal positions. If a country is positioned more centrally in the trade network, it signifies a stronger ability to control and access resources and information.

(5) It is suggested to focus on the impact of ICT, network infrastructure, informatization and other economic variables on GVC in the literature review, the existing references pay less attention to the core issues of this paper

Response: Thank you for your specific and professional comments and suggestions. We incorporated the impacts of information and communication technology, network infrastructure, informatization, and other economic variables on the global value chain in our literature review. The specific modifications are as follows:

Additionally, research suggests that information and communication technology (ICT) plays a significant role in driving global value chain (GVC) activities[19]. Studies from both macro and micro perspectives have found that the development of the internet notably promotes GVC trade[20]. This promotion is mainly achieved by enhancing supply-demand matching and fostering communication and collaboration, injecting new vitality into the global value chain[21]. However, there are also opinions suggesting that ICT may have adverse effects on the upgrading of GVCs. Under the global value chain division system driven by labor arbitrage, the substitution effect of automation on labor leads to the contraction of GVC trade[22].

19. Freund C., Weinhold D. The Internet and International Trade in Services[J]. American Economic Review, 2002, 92(2):236-240.

20. Xing Z W. The Impacts of Information and Communications Technology (ICT) and E-commerce on Bilateral Trade Flows[J]. International Economics and Economic Policy, 2018, 15(3):565-586.

21. Antràs P. Conceptual Aspects of Global Value Chains[J]. World Bank Economic Review, 2020, 34(3):551-574.

22. Artuc E., Bastos P., Rijkers B. Robots, Tasks and Trade[R]. Policy Research Working Paper, 2018, No.8674.

Finally, we sincerely thank you again for reviewing this manuscript and providing us with valuable comments and suggestions for revising and improving the manuscript. We look forward to hearing from you in due time regarding this submission and to respond to any further questions and comments you may have.

Yours sincerely

The Authors

Apr 2024

---

## [Decision Letter · Decision Letter 2]

PONE-D-23-10130R2Digital Economy Development and Global Value Chain Network CentralizationPLOS ONE

Dear Dr. yue,

Thank you for submitting your manuscript to PLOS ONE. After careful consideration, we feel that it has merit but does not fully meet PLOS ONE’s publication criteria as it currently stands. Therefore, we invite you to submit a revised version of the manuscript that addresses the points raised during the review process.

For detailed comments, please refer to the end of this email. Please pay attention to the recommendations of Reviewer 5 and provide relevant modifications. 

We look forward to receiving your revised manuscript.

Kind regards,

Imran Ur Rahman, Ph.D

Academic Editor

PLOS ONE

**Additional Editor Comments:**

I appreciate the efforts of the authors in the revision of the manuscript. Although most of the comments have been addressed, there are still some minor points that need to be revised.

1. Please enhance the literature further and provide more relevant references for some contemporary literary works.

2. There are minor grammatical errors. The authors should revise the language and formatting of the article.

Reviewers' comments:

Reviewer's Responses to Questions

**Comments to the Author**

1. If the authors have adequately addressed your comments raised in a previous round of review and you feel that this manuscript is now acceptable for publication, you may indicate that here to bypass the “Comments to the Author” section, enter your conflict of interest statement in the “Confidential to Editor” section, and submit your "Accept" recommendation.

Reviewer #3: All comments have been addressed

Reviewer #5: (No Response)

2. Is the manuscript technically sound, and do the data support the conclusions?

Reviewer #3: Yes

Reviewer #5: Partly

3. Has the statistical analysis been performed appropriately and rigorously? 

Reviewer #3: Yes

Reviewer #5: No

4. Have the authors made all data underlying the findings in their manuscript fully available?

Reviewer #3: Yes

Reviewer #5: No

5. Is the manuscript presented in an intelligible fashion and written in standard English?

Reviewer #3: Yes

Reviewer #5: Yes

6. Review Comments to the Author

Reviewer #3: (No Response)

Reviewer #5: Overview:

The authors propose a macroeconomic model to study the impact of the digital economy on the global value chain (GVC). They then empirically test how a measured index of GVC participation relates to industry estimates of the digital economy and other macro variables, using fixed effects.

General Comments:

The inclusion of the “digital economy” in the model feels oversimplified. The model does not seem to adequately capture the unique ways in which the digital economy affects the broader economy. Furthermore, the authors should avoid claiming causality from their empirical study. While fixed effects can reduce bias, they are not sufficient for establishing causality.

Regarding the Theory:

The authors depart from the basic 2-product CES production function by adding another production input. However, what is the contribution of this addition? What makes the digital economy so special and unique? Is the digital economy an actual factor of production, or is it more of an intermediate product? A simpler 2-good model, where the digital economy acts as an intermediate good used to produce a final product for exports, might have been more convincing. However, similar models (with intermediate goods) have been explored in other contexts (e.g., Caliendo and Parro, 2015), leaving the question of what makes the digital economy unique unanswered.

Regarding the Empirical Analysis:

At the macroeconomic level, establishing causality is challenging, if not impossible, at least in its contemporary sense. This limitation is not the authors' fault, but it should be addressed in the study. While it is valid to empirically test the relationships derived from a theoretical model, the text should acknowledge this limitation. The authors should neither state nor imply causality. For instance, the second sentence of the abstract reads: “this paper […] empirically examines the impact of digital economy development on the centralization of the GVC network and the internal mechanism.” There are several other instances where causality is either stated or implied.

Minor Comments:

There are several typos. For example, on page 11, line 101, “mentined” should be “mentioned.”

There are also several punctuation errors, such as a missing space after “jobs” on page 11, line 96, and missing spaces between $alpha$ and $\eta$ on page 13, line 184.

The mathematical notation contains inaccuracies. For instance, on page 13, line 183, $q_i$ is not a set but a heterogeneous good. Additionally, the way the integral is written implies that $\Omega$ is a set of indexes $i$, not a set of goods.

Quoting Melitz and Ottaviano (2008), “the parameter $\gamma$ indexes the degree of product differentiation between the varieties,” which is neither defined as an elasticity nor “happens to be equal” to an elasticity. Since the demands are linear (due to quadratic utility), the elasticities are definitely not constant.

References

Caliendo, L., \& Parro, F. (2015). Estimates of the Trade and Welfare Effects of NAFTA. The Review of Economic Studies, 82(1), 1-44.

Melitz, M. J., \& Ottaviano, G. I. (2008). Market size, trade, and productivity. The Review of Economic Studies, 75(1), 295-316.

7. PLOS authors have the option to publish the peer review history of their article (what does this mean? ). If published, this will include your full peer review and any attached files.

If you choose “no”, your identity will remain anonymous, but your review may still be made public.

**Do you want your identity to be public for this peer review?** For information about this choice, including consent withdrawal, please see our Privacy Policy .

Reviewer #3: No

Reviewer #5: No

---

## [Author Response · Author response to Decision Letter 3]

15 Sep 2024

Response to Editor and Reviewers

Dear Editor and Reviewers:

First of all, please allow us to take this opportunity to express our heartfelt thanks to you for taking time out from your busy schedule to review this manuscript. You have provided us with constructive comments and suggestions, which are of great help for us to further improve this manuscript. We have carefully reviewed and revised the manuscript according to your valuable comments and suggestions. Here, we explain the revised work in detail below and provide the point-by-point responses to the editor’s and reviewers’ comments. Moreover, we have included continuous line numbers in the revised manuscript and have marked all variations in red.

Response to Editor:

(1) Please enhance the literature further and provide more relevant references for some contemporary literary works.

Response: Thank you for your specific and professional comments and suggestions. The authors followed the reviewer's comments and added appropriate literature to enhance the scholarly nature of the paper, which included:

Caliendo, L., & Parro, F. (2015). Estimates of the Trade and Welfare Effects of NAFTA. The Review of Economic Studies, 82(1), 1-44.

Melitz, M. J., & Ottaviano, G. I. (2008). Market size, trade, and productivity. The Review of Economic Studies, 75(1), 295-316.

(2) There are minor grammatical errors. The authors should revise the language and formatting of the article.

Response: Thank you for your specific and professional comments and suggestions. The corresponding grammar and expression have been modified in this paper, specifically as follows:

Most of the studies mentioned above have focused on the impact of the digital economy on the "binary margin" of firms' exports,�page 11, line 101

but it also has a substitution effect on the traditional labor force in terms of creating new jobs�page 11, line 96

α and η represent the elasticity of substitution between homogeneous and heterogeneous products�page 13, line 184.

q_(i ) represents a heterogeneous product�page 13, line 183

Ω is a set of indexes�page 13, line 185

γ represents the degree of product differentiation between the varieties �page 13, line 183

Response to Reviewer :

(1) Regarding the Theory:

The authors depart from the basic 2-product CES production function by adding another production input. However, what is the contribution of this addition? What makes the digital economy so special and unique? Is the digital economy an actual factor of production, or is it more of an intermediate product? A simpler 2-good model, where the digital economy acts as an intermediate good used to produce a final product for exports, might have been more convincing. However, similar models (with intermediate goods) have been explored in other contexts (e.g., Caliendo and Parro, 2015), leaving the question of what makes the digital economy unique unanswered.

Response: Thank you for your specific and professional comments and suggestions. In the original model, this paper regards the digital economy as a production factor input into the production of firms. As the reviewer said, it is more appropriate to put the digital economy into production as an intermediate. Therefore, this paper refers to the practice of Caliendo and Parro (2015) and puts the digital economy into production as an intermediate. But the authors also want to emphasize that whether the digital economy is put into production as a factor of production or an intermediate, the conclusion that the development of the digital economy affects the centralization of the global value chain network by lowering the price of digital factors will not change. Therefore, the core conclusion of this paper will not change. Specific modifications to the model are as follows:

According to the practice of Caliendo and Parro, the firm production function is set as follows:

y(φ)=z(φ)[l(φ)]^(γ^l ) [s(φ)]^(γ^s ) ∏_(k=1)^n▒[m(φ)]^(γ^k ) �5

Equations (5) represents the final product produced by the enterprise is composed of a series of intermediate products, z(φ) represents the productivity of the firm. l(φ) represents labor input. s(φ) represents digital technology. m(φ) represents the input of a series of other intermediate goods. γ^l, γ^s, and γ^k represent the input shares of labor, digital technology, and other intermediate goods, respectively. According to the principle of firm cost minimization, the firm's marginal cost is:

c=Υw^(γ^l ) p_s^(γ^s ) ∏_(k=1)^n▒〖(p_m)〗^(γ^k ) �6

Where Υ=(γ^l )^(-γ^l ) (γ^s )^(-γ^s ) ∏_(k=1)^n▒〖(γ^k)〗^(γ^(-k) ) . w, p_s and p_m represent wage price, digital technology price and other intermediate goods price respectively.

(2) Regarding the Empirical Analysis:

At the macroeconomic level, establishing causality is challenging, if not impossible, at least in its contemporary sense. This limitation is not the authors' fault, but it should be addressed in the study. While it is valid to empirically test the relationships derived from a theoretical model, the text should acknowledge this limitation. The authors should neither state nor imply causality. For instance, the second sentence of the abstract reads: “this paper […] empirically examines the impact of digital economy development on the centralization of the GVC network and the internal mechanism.” There are several other instances where causality is either stated or implied.

Response: Thank you for your specific and professional comments and suggestions. This paper should not state or imply causality between macro variables in the abstract. Therefore, the paper describes the relationship between the two variables in the abstract as:

empirically examines the effects of digital economy development on the centralization of the GVC network and the internal mechanisms.

(3) Minor Comments:

There are several typos. For example, on page 11, line 101, “mentined” should be “mentioned.”

There are also several punctuation errors, such as a missing space after “jobs” on page 11, line 96, and missing spaces between $alpha$ and $\eta$ on page 13, line 184.

The mathematical notation contains inaccuracies. For instance, on page 13, line 183, $q_i$ is not a set but a heterogeneous good. Additionally, the way the integral is written implies that $\Omega$ is a set of indexes $i$, not a set of goods.

Quoting Melitz and Ottaviano (2008), “the parameter $\gamma$ indexes the degree of product differentiation between the varieties,” which is neither defined as an elasticity nor “happens to be equal” to an elasticity. Since the demands are linear (due to quadratic utility), the elasticities are definitely not constant.

Response: Thank you for your specific and professional comments and suggestions. The corresponding grammar and expression have been modified in this paper, specifically as follows:

Most of the studies mentioned above have focused on the impact of the digital economy on the "binary margin" of firms' exports,�page 11, line 101

but it also has a substitution effect on the traditional labor force in terms of creating new jobs�page 11, line 96

α and η represent the elasticity of substitution between homogeneous and heterogeneous products�page 13, line 184.

q_(i ) represents a heterogeneous product�page 13, line 183

Ω is a set of indexes�page 13, line 185

γ represents the degree of product differentiation between the varieties �page 13, line 183

Finally, we sincerely thank you again for reviewing this manuscript and providing us with valuable comments and suggestions for revising and improving the manuscript. We look forward to hearing from you in due time regarding this submission and to respond to any further questions and comments you may have.

Yours sincerely

The Authors

Sep 2024

---

## [Decision Letter · Decision Letter 3]

PONE-D-23-10130R3Digital Economy Development and Global Value Chain Network CentralizationPLOS ONE

Dear Dr. yue,

Thank you for submitting your manuscript to PLOS ONE. After careful consideration, we feel that it has merit but does not fully meet PLOS ONE’s publication criteria as it currently stands. Therefore, we invite you to submit a revised version of the manuscript that addresses the points raised during the review process.

We look forward to receiving your revised manuscript.

Kind regards,

Imran Ur Rahman, Ph.D

Academic Editor

PLOS ONE

Additional Editor Comments:

Authors are unable to provide adequate response to the concerns and comments of reviewer 5.

Reviewers' comments:

Reviewer's Responses to Questions

**Comments to the Author**

1. If the authors have adequately addressed your comments raised in a previous round of review and you feel that this manuscript is now acceptable for publication, you may indicate that here to bypass the “Comments to the Author” section, enter your conflict of interest statement in the “Confidential to Editor” section, and submit your "Accept" recommendation.

Reviewer #5: (No Response)

2. Is the manuscript technically sound, and do the data support the conclusions?

Reviewer #5: No

3. Has the statistical analysis been performed appropriately and rigorously? 

Reviewer #5: No

4. Have the authors made all data underlying the findings in their manuscript fully available?

Reviewer #5: No

5. Is the manuscript presented in an intelligible fashion and written in standard English?

Reviewer #5: Yes

6. Review Comments to the Author

Reviewer #5: In my previous review, I wrote: “I do not like the oversimplified way the ‘digital economy’ was incorporated into the model. I do not see how this model captures the specific manner in which the digital economy affects the broader economy. Moreover, the authors should not claim causality from their empirical study. While fixed effects may reduce bias, they are not sufficient for identification purposes.” Unfortunately, none of these concerns have been addressed.

Regarding Theory:

I previously referenced Caliendo and Parro (2015), stating: “I would have been more convinced by a simpler two-good model where the ‘digital economy’ is treated as an intermediate product used to produce a final good for export.” I did not suggest that the authors adopt the Caliendo and Parro (2015) model or simply relabel variables in an existing framework. In fact, I followed up by stating: “However, such models have already been studied in other contexts […] and the question of ‘what makes the digital economy special and unique?’ remains unanswered.”

In other words, I emphasized that an original research paper in macroeconomic theory should propose a model that explicitly demonstrates how the digital economy differs from other inputs or intermediate goods, thereby adding new insights to the literature. Instead, the authors appear content with adapting existing models by simply renaming variables, resulting in a theoretical contribution that is minimal or even absent.

Regarding Empirics:

I previously provided specific examples illustrating how the authors claim causality without properly establishing it with their data. While my earlier examples focused on the abstract for conciseness, the same issue persists throughout the paper. Despite this, the authors only changed the abstract.

As I mentioned in my earlier review: “The authors are empirically testing relationships derived from a theoretical model. There is nothing wrong with this approach, but the text should acknowledge its limitations: the authors should neither claim nor imply causality. For example, the second sentence of the abstract reads…” Although the abstract has been revised, the remainder of the paper continues to imply causality. For instance, section 5.2 is devoted to discussing endogeneity.

The authors’ empirical analysis merely shows that their measure of the digital economy is positively correlated with the centralization of the value chain network. It does not demonstrate that an increase in the digital economy unambiguously leads to a $\beta$ increase in centralization.

Conclusion

While I acknowledge that the authors have conducted some work and produced certain results, their scientific contribution may be a better fit for more specialized journals, rather than a general interest journal such as PLOS ONE.

References

Caliendo, L., and Parro, F. (2015). Estimates of the Trade and Welfare Effects of NAFTA. The Review of Economic Studies, 82(1), 1-44.

7. PLOS authors have the option to publish the peer review history of their article (what does this mean? ). If published, this will include your full peer review and any attached files.

**Do you want your identity to be public for this peer review?** For information about this choice, including consent withdrawal, please see our Privacy Policy .

Reviewer #5: No

---

## [Author Response · Author response to Decision Letter 4]

23 Dec 2024

Response to Editor and Reviewers

Dear Editor and Reviewers:

First of all, please allow us to take this opportunity to express our heartfelt thanks to you for taking time out from your busy schedule to review this manuscript. You have provided us with constructive comments and suggestions, which are of great help for us to further improve this manuscript. We have carefully reviewed and revised the manuscript according to your valuable comments and suggestions. Here, we explain the revised work in detail below and provide the point-by-point responses to the editor’s and reviewers’ comments. Moreover, we have included continuous line numbers in the revised manuscript and have marked all variations in red.

In response to the comments raised by the reviewers, the authors have made detailed revisions. Firstly, we would like to briefly report to the review experts the main content of this revision: The revision primarily focuses on the theoretical model and empirical research, particularly the discussion on causality. In the revision of the theoretical model, data elements in the digital economy are mainly introduced as production factors into enterprises' production decisions, influencing their operational performance. This emphasizes the unique position and role of the digital economy within the model. In the revision of the empirical research, we have conducted supplementary discussions on the causality between the development of the digital economy and the centralization of global value chain networks. This includes adding correlation coefficient tests and employing the instrumental variable approach to address potential endogeneity issues. However, the authors also acknowledge that their discussion on causality is only based on our best efforts to analyze, and there is certainly room for improvement. Therefore, further research on the relationship between the development of the digital economy and the centralization of global value chain networks will be a direction for the authors' future endeavors. Thank you for the understanding and inclusiveness.

Response to Reviewer :

1�Regarding the Theory:

Reviewer’s comment: I emphasized that an original research paper in macroeconomic theory should propose a model that explicitly demonstrates how the digital economy differs from other inputs or intermediate goods, thereby adding new insights to the literature. Instead, the authors appear content with adapting existing models by simply renaming variables, resulting in a theoretical contribution that is minimal or even absent.

Authors’ response: We appreciate the reviewer's comments and have revised the theoretical section accordingly. In particular, we have emphasized the distinction between the digital economy and other economic factors, as well as their specific impacts on overall corporate production decisions. Therefore, in the revised theoretical model, we have retained the basic framework of Melitz & Ottaviano (2008) and extended the endogenous growth model of De Ridder (2021) to construct a theoretical framework for corporate export behavior incorporating data as a factor of production.

The most significant modification in our model is the inclusion of data as a production factor in corporate production decisions, which further influences corporate operating performance. This underscores the unique role and importance of the digital economy within the model.

Why can data within the digital economy serve as a production factor influencing corporate production decisions? The reason is that the digital economy relies on data as a key production factor, with digital technology as its core driving force and modern information networks as its important carrier. Through the deep integration of digital technology with the real economy, it continuously enhances the digitization, networking, and intelligence levels of the economy and society, accelerating the reconstruction of economic development and governance models. Data is a crucial production factor in the digital economy era, often compared to coal and oil in the digital age. Firstly, as a new input, data can drive economic growth on its own. Unlike traditional production factors such as capital and labor, data has characteristics such as extremely low search costs, zero replication costs, extremely low transmission costs, and non-excludability (Goldfarb & Tucker, 2019). Data not only has nearly infinite supply but also has a wide range of applications, thus being able to break through the growth constraints of traditional production factors. Mainstream economic growth models have begun to introduce variables related to the digital economy, such as data capital and robots (Acemoglu & Restrepo, 2018). Secondly, digital technology enhances capital and labor factors, increasing their marginal output. From a capital perspective, the diffusion of digital technology can reshape a firm's production and management methods, enhancing the efficiency of capital accumulation. Therefore, in the current digital economy era, data as a key factor is a fundamental characteristic of the digital economy, continuously influencing corporate production decisions.

In the producer model, we assume that firms require labor input (l_i) and data capital (f_i) to produce goods. Since big data enables firms to make more scientific production decisions, we assume that data capital investment can reduce firms' marginal costs (s_i). That is, c_i=s_i ω, where s_i∈[0,1], ω stands for wage level Hence, the production function of the firm can be expressed as:

y_i=l_i/s_i �5

The combination of data capital and labor has two characteristics in terms of its impact on output: 1. There are differences in the efficiency of different firms in developing and utilizing data capital, φ_i represented by firm-specific cost parameters. Due to the economies of scale and scope of data factors, cost parameters φ_i significantly decrease with increased data capital investment f_i. 2. Firms invest in data capital before observing their competitors' marginal costs and pricing. When firms set prices and make production decisions, data assets have already become sunk costs.

The functional relationship between a firm's data capital f_i and the decrease in marginal cost s_i is given by f(s_i�φ_i ). The function f_i is twice differentiable, strictly decreasing when s_i∈[0,1], and increasing when φ_i>0. It is assumed that f(s_i�φ_i ) is convex and decreasing in s_i, and satisfies f(1�φ_i )=0�lim┬(s_i→0)⁡〖f(s_i�φ_i )=∞〗. For the purpose of analysis, this paper selects an exponential function form for the data capital function that meets these characteristics:

f(s_i�φ_i )=φ_i ([1/s_i ]^ψ-1) �6

Where ψ>0 represents the curvature parameter.

Subsequently, the decision-making process for the firm unfolds as follows: First, the firm decides on its data capital expenditure f_i, that is, whether to produce at a marginal cost lower than the benchmark ω; then, the firm select s_i and incurs the corresponding data capital expenditure f_i; finally, the firm observes its competitors' marginal costs and sets its price p_i. The analysis will proceed using backward induction.

Decision on Data Capital Investment: Combining Equation (4), the s_i that minimizes marginal cost at the optimal output y_i^* is:

s_i^*=min[([y_i^* ]^(-1) ψφ_i )^(1/(ψ+1))�1] �7

The modeling approach for producer behavior can be summarized as follows: With the development of the digital economy, data elements f_i are invested as important production factors in enterprise production, and as the digital economy becomes more developed, the marginal cost s_i of enterprises decreases.

(2) Regarding the Empirical Analysis:

Reviewer’s comment: "The authors are empirically testing relationships derived from a theoretical model. There is nothing wrong with this approach, but the text should acknowledge its limitations: the authors should neither claim nor imply causality. For example, the second sentence of the abstract reads…" Although the abstract has been revised, the remainder of the paper continues to imply causality. For instance, section 5.2 is devoted to discussing endogeneity.

Authors’ response: We appreciate the reviewer's constructive comments and respond to the reviewer's concerns as follows: As the reviewer mentioned, the reviewer's questions primarily focus on the causality between the two variables selected in this paper, namely, whether the development of the digital economy leads to centralization in global value chain networks.

We clarify our writing approach: Firstly, in the theoretical analysis, we established a heterogeneous firm model and constructed a theoretical framework for corporate export behavior incorporating data as a factor of production. In the theoretical section, we found that the development of the digital economy enhances a country's central position in global value chain networks. Subsequently, this paper empirically tests the conclusions of the theoretical analysis. In the empirical tests, we first supplemented the correlation coefficient test, with the results shown in Table 2, revealing a positive correlation between the development of the digital economy and centralization in global value chain networks. Then, we proceed to assess the causality between the development of the digital economy and centralization in global value chain networks.

Table 2. Correlation coefficient test.

centralization Digit Labor Persons Scale Fdi GDP RTA

centralization 1.000

Digit 0.056*** 1.000

Labor 0.162*** 0.002 1.000

Persons -0.072*** -0.001 -0.364*** 1.000

Scale 0.100*** 0.000 0.077*** 0.058*** 1.000

Fdi -0.025** 0.066*** 0.127*** 0.000 -0.325*** 1.000

GDP 0.432*** -0.002 0.343*** 0.163*** 0.207*** 0.024** 1.000

RTA 0.099*** 0.013 0.387*** 0.183*** 0.391*** 0.303*** 0.276 1.000

As the reviewer mentioned, judging causality is one of the difficulties in current economic research, and the biggest challenge in judging the causality between two variables lies in the issue of endogeneity. Endogeneity mainly manifests as omitted variables and reverse causality. In the baseline regression tests, we have added control variables as much as possible and found that whether control variables are included or not, the coefficients and significance of the core explanatory variables do not change significantly, indicating that the issue of omitted variables in this paper is relatively minor. Therefore, the main challenge in determining the causality between the development of the digital economy and centralization in global value chain networks lies in addressing reverse causality. The primary method in current economic research to address reverse causality is the use of instrumental variables. In our original research, we referred to the study by Nunn & Qian and used the number of telephone lines worldwide in 1984 as an instrumental variable for the development of the digital economy. The conclusions in Table 4 support the baseline regression results.

However, to make our research results more reliable and the causality between the development of the digital economy and centralization in global value chain networks more credible, in the revised manuscript, we have selected another instrumental variable to address potential endogeneity issues. Reference to Beverelli et al. (2017), we constructed a weighted value of the digital economy development level of other countries relative to the home country and the average digital economy development level of countries at different income levels, multiplied by the digitalization rate of various industries in each country, serving as an instrumental variable for the digital economy at the "country-industry" level. The rationale for this "country-industry" level digital economy instrumental variable is as follows: In terms of relevance, countries with similar economic development levels tend to have similar levels of digital economy development, satisfying the relevance requirement. In terms of exogeneity, the average level of digital economy development across different income levels reflects the digital economy development within that income bracket but does not contain country-specific development characteristics. Therefore, this instrumental variable meets the requirements. Table 5 shows the regression results using the "country-industry" level instrumental variable, indicating that after addressing potential endogeneity issues, the conclusion that the development of the digital economy enhances centralization in global value chain networks still holds.

Table 5. Regression results of instrumental variables at the "country-industry" level.

1� �2

2SLS 2SLS

Digit 0.0987***(0.0131) 0.0881***(0.0134)

Control variables NO YES

Country fixed effects YES YES

Time fixed effects YES YES

Industry - year fixed effects YES YES

Kleibergen-Paap rk LM Statistics 2.8e+06***[0.000] 2.7e+06***[0.000]

Cragg-Donald Wald F Statistics 3.2e+06{16.38} 3.4e+06{16.38}

Kleibergen-Paap rk Wald FStatistics 2.7e+06{16.38} 2.8e+06{16.38}

N 9625 8818

Through the above series of tests, we have basically concluded that the development of the digital economy enhances centralization in global value chain networks. However, as the reviewer pointed out, verifying the causality between two variables is a difficult task in economic research, and we have also made every effort to study the impact of the development of the digital economy on centralization in global value chain networks. We acknowledge that there is room for improvement in our current research, which is also a direction we will strive towards in the future. We thank the reviewer for your understanding and consideration.

Finally, we sincerely thank you again for reviewing this manuscript and providing us with valuable comments and suggestions for revising and improving the manuscript. We look forward to hearing from you in due time regarding this submission and to respond to any further questions and comments you may have.

Yours sincerely

The Authors

Dec 2024

---

## [Decision Letter · Decision Letter 4]

PONE-D-23-10130R4Digital Economy Development and Global Value Chain Network CentralizationPLOS ONE

Dear Dr. yue,

Thank you for submitting your manuscript to PLOS ONE. After careful consideration, we feel that it has merit but does not fully meet PLOS ONE’s publication criteria as it currently stands. Therefore, we invite you to submit a revised version of the manuscript that addresses the points raised during the review process.

For detailed comments, please check the recommendations provided at the end of this emails. 

We look forward to receiving your revised manuscript.

Kind regards,

Imran Ur Rahman, Ph.D

Academic Editor

PLOS ONE

Journal Requirements:

Additional Editor Comments:

This study addresses a timely and important topic—the role of digital economy development in shaping a country’s position in global value chains (GVCs). This paper makes a valuable empirical contribution but needs key points to be revised:

1. Please enhance the title of the study. In the abstract, replace "high-quality opening up" with standard terms (e.g., "economic liberalization").

2. The literature review needs to be enhanced, and more up-to-date literature studies should be added.

3. In the result section, please explain if the Hausman test was used to determine the model specification and add the chi-squared value in the table.

4. What does UIBE stand for? Please add it for the clarity of the readers.

5. Please further justify the use of the time frame 2004-2014. Why not till 2024 or 2023?

6. Further explain the selection and number of countries used and provide a list of the selected countries in the appendix.

7. If possible, add a mechanism diagram or the research theoretical framework.

8. Replace or rephrase vague phrases and overstated claims throughout the manuscript.

Reviewers' comments:

Reviewer's Responses to Questions

**Comments to the Author**

1. If the authors have adequately addressed your comments raised in a previous round of review and you feel that this manuscript is now acceptable for publication, you may indicate that here to bypass the “Comments to the Author” section, enter your conflict of interest statement in the “Confidential to Editor” section, and submit your "Accept" recommendation.

Reviewer #6: (No Response)

Reviewer #7: All comments have been addressed

2. Is the manuscript technically sound, and do the data support the conclusions?

Reviewer #6: Yes

Reviewer #7: Yes

3. Has the statistical analysis been performed appropriately and rigorously? 

Reviewer #6: Yes

Reviewer #7: Yes

4. Have the authors made all data underlying the findings in their manuscript fully available?

Reviewer #6: Yes

Reviewer #7: Yes

5. Is the manuscript presented in an intelligible fashion and written in standard English?

Reviewer #6: Yes

Reviewer #7: Yes

6. Review Comments to the Author

Reviewer #6: 1. Clarify Theoretical Contributions

• The paper extends existing models (Melitz & Ottaviano, De Ridder) but could more explicitly highlight how its contributions differ from prior work.

• Clearly articulate how introducing data as a production factor changes the implications for global value chain (GVC) centralization.

2. Strengthen Causality Claims

• The study aims to address causality concerns, but there are still implied causal claims in certain sections (e.g., “The development of the digital economy significantly enhances a country's central location in the GVC network”).

• While instrumental variables (IV) were used, discussing potential limitations of the IV approach would strengthen credibility.

3. Improve Flow and Readability

• Some sections, especially the methodology and theoretical model, are dense with mathematical formulations. Consider adding brief explanations or intuitive interpretations before presenting equations.

• The literature review is comprehensive but could be structured better—perhaps by grouping studies into broader themes (e.g., digital economy impact on firms, trade networks, productivity).

4. Refine Abstract and Introduction

• The abstract is clear but could be more concise in summarizing key findings.

• The introduction could benefit from a stronger statement of research gaps and practical significance.

5. Ensure Consistent Terminology

• Terms like "centralization of GVC networks" and "central location in the GVC" should be consistently defined and used throughout.

Reviewer #7: Overall Evaluation: This article explores the impact of the digital economy on the centralization of global value chain networks. The article boasts a clear structure, rigorous logic, a well-constructed theoretical model, and solid empirical analysis, leading to conclusions with significant policy implications. Specifically, by introducing factors related to digital economy development, the article extends the heterogeneous firm trade model and conducts an in-depth analysis of the impact mechanism of the digital economy on the centralization of global value chain networks from both theoretical and empirical perspectives. Moreover, the article employs various methods to address endogenous issues and conducts robustness tests, ensuring the reliability and validity of the research results. Overall, this is a paper with high academic value and practical significance. However, there is still room for improvement in the following aspects:

Suggestion for Revision:

1.In the introduction section, it can be further clarified to explicitly state the motivation for studying the impact of the digital economy on the centralization of global value chain networks. This should emphasize how, against the current global economic backdrop, the digital economy has emerged as a crucial factor in driving the enhancement of countries' positions within global value chains.

2. In the data description section, the data in the text is up to 2014. It is suggested that the limitations of the data be mentioned in the discussion and that the updated data can be combined in the future to verify the sustainability of the conclusion.

3. The robustness test section can be appropriately expanded to supplement and replace the robustness tests of samples (such as excluding countries or industries with extreme values).

4. The conclusions need to be more targeted. The current suggestions are relatively macro-level. It is recommended to refine the policy directions based on the research findings. For instance, for developing countries: propose strengthening digital infrastructure construction (such as 5G networks) and promoting labor skill training to address automation substitution; for industries with high institutional quality: suggest optimizing the intellectual property protection system to promote the deep integration of digital technology and manufacturing.

7. PLOS authors have the option to publish the peer review history of their article (what does this mean? ). If published, this will include your full peer review and any attached files.

**Do you want your identity to be public for this peer review?** For information about this choice, including consent withdrawal, please see our Privacy Policy .

Reviewer #6: No

Reviewer #7: No

---

## [Author Response · Author response to Decision Letter 5]

20 May 2025

Response to Editor and Reviewers

Dear Editor and Reviewers:

First of all, please allow us to take this opportunity to express our heartfelt thanks to you for taking time out from your busy schedule to review this manuscript. You have provided us with constructive comments and suggestions, which are of great help for us to further improve this manuscript. We have carefully reviewed and revised the manuscript according to your valuable comments and suggestions. Here, we explain the revised work in detail below and provide the point-by-point responses to the editor’s and reviewers’ comments. Moreover, we have included continuous line numbers in the revised manuscript and have marked all variations in red.

Response to Editor:

(1) Please enhance the title of the study. In the abstract, replace "high-quality opening up" with standard terms (e.g., "economic liberalization").

Response: Thank you for your specific and professional comments and suggestions. We have made corresponding revisions to the wording in the abstract, replacing "high-quality opening up" with "economic liberalization".

(2) The literature review needs to be enhanced, and more up-to-date literature studies should be added.

Response: Thank you for your specific and professional comments and suggestions. We have incorporated the latest literature in the literature review section of the revised manuscript. The newly added references encompass:

44. Fan Z , Zhou Y , Anwar S .Centralization of trade agreements network and global value chain participation[J].Quarterly Review of Economics and Finance, 2024, 94:11-24.

45. Li H , Yang Z .Does digital economy development affect urban environment quality: Evidence from 285 cities in China[J].PLoS ONE, 2024, 19(2):26.

(3) In the result section, please explain if the Hausman test was used to determine the model specification and add the chi-squared value in the table.

Response: Thank you for your specific and professional comments and suggestions. We have supplemented the results of the Hausman test in the revised manuscript. The results show that the P-value of the Hausman test is less than 0.01, leading to the rejection of the null hypothesis of random effects. Therefore, a fixed effects regression model should be selected for the analysis.

Table 3. Baseline regression results of the development of the digital economy on the centralization of the value chain network.

1� �2� �3� �4

FE FE FE FE

Digit 0.0837***(0.0294) 0.1990***(0.0342) 0.0857***(0.0310) 0.0876***(0.0325)

Labor 0.0018(0.0073) -0.0016(0.0069) -0.0015(0.0067)

Persons -2.5502***(0.5741) -2.1943***(0.5575) -2.2008***(0.5537)

Scale 1.1061***(0.3982) 1.2752***(0.3740) 1.2734***(0.3712)

Fdi -0.0017(0.0035) -0.0031(0.0032) -0.0031(0.0032)

GDP 0.0814**(0.0330) 0.1022***(0.0345) 0.1024***(0.0348)

RTA 0.0448(0.0328) 0.0687**(0.0321) 0.0683**(0.0338)

Constant term 6.688***(0.0682) 15.142***(2.6211) 13.293***(2.5762) 13.324***(2.5692)

Country fixed effects YES YES YES YES

Industry fixed effects YES NO YES NO

Hausman Test 13.54�p=0.0003� 14.89�p=0.0001� 18.99�p=0.000� 12.06�p=0.0006

N 9625 8818 8818 8818

R2 0.632 0.380 0.630 0.632

(4) What does UIBE stand for? Please add it for the clarity of the readers.

Response: Thank you for your specific and professional comments and suggestions. We have clarified the meaning of "UIBE" (University of International Business and Economics) at its first occurrence in the revised manuscript. Additionally, we have provided a detailed description of the "UIBE GVC Indicators database" in the "Data description" section under 4.3.1.

(2) UIBE GVC Indicators database

The UIBE GVC Indicators Database is an open-access database constructed and maintained by the Global Value Chain Institute at the University of International Business and Economics (UIBE). This database is designed to provide useful indicators and data support for international trade research and global value chain (GVC) analysis.The UIBE GVC Indicators database is based on the input‒output data of countries and regions in the world and utilizes the structural decomposition method of input‒output to calculate the indicators of value-added trade and GVC participation, which are used to calculate the core explanatory variables of this paper[2].

(5) Please further justify the use of the time frame 2004-2014. Why not till 2024 or 2023?

Response: Thank you for your specific and professional comments and suggestions. The reason the sample period in this study only extends up to 2014 is due to the limitations of the database used. The calculation of the core variables in this paper relies on the WIOD (World Input-Output Database), which, as of now, has only been updated up to the year 2014. Furthermore, the data from 2004 to 2014 used in this study are sufficient to essentially demonstrate the conclusions of our research questions. Of course, once the WIOD database is updated in the future, we will also update the sample data for subsequent research. We hope the editor can understand this limitation.

(6) Further explain the selection and number of countries used and provide a list of the selected countries in the appendix.

Response: Thank you for your specific and professional comments and suggestions. In the appendix section of the revised manuscript, we have provided a detailed explanation of the data on the number of countries used in our study and listed the specific names of these countries.

Appendix

The 2016 version of the WIOD (World Input-Output Database) includes a total of 42 countries, which are as follows: Australia、Austria、Belgium、Bulgaria、Brazil、Canada、Switzerland、China、Cyprus、Czech Republic、Germany、Denmark、Spain、Estonia、Finland、France、United Kingdom、Greece、Croatia、Hungary、Indonesia、India、Ireland、Italy、Japan、Korea、Lithuania、Luxembourg、Latvia、Mexico、Malta、Netherlands、Norway、Poland、Portugal、Romania、Russia、Slovakia、Slovenia、Sweden、Turkey、United States.

(7) If possible, add a mechanism diagram or the research theoretical framework.

Response: Thank you for your specific and professional comments and suggestions. We have supplemented and drawn the theoretical framework diagram in the revised manuscript.

Fig. 1 A Diagram Illustrating the Mechanism by Which the Development of the Digital Economy Influences the Centralization of Global Value Chain Network

(8) Replace or rephrase vague phrases and overstated claims throughout the manuscript.

Response: Thank you for your specific and professional comments and suggestions. We have revised the expressions in this paper in the revised manuscript, removing vague phrases and overstated claims.

Response to Reviewer6 :

(1) Clarify Theoretical Contributions

The paper extends existing models (Melitz & Ottaviano, De Ridder) but could more explicitly highlight how its contributions differ from prior work.

Clearly articulate how introducing data as a production factor changes the implications for global value chain (GVC) centralization.

Response: Thank you for your specific and professional comments and suggestions. Firstly, regarding the theoretical contributions of this paper, our theoretical framework is derived from the Melitz & Ottaviano model, yet it distinctly differs from it. The most significant contribution lies in the expansion of the Producer behavior aspect. The Melitz & Ottaviano model primarily assumes that firms produce using only labor as a factor of production.. In contrast, this paper argues that in addition to the traditional factor of labor, data capital is also a crucial factor of production. This contribution not only extends the Producer behavior in the Melitz & Ottaviano model but also lays a theoretical foundation for our analysis of how the development of the digital economy influences the centralization of global value chain networks.

Subsequently, in the revised manuscript, we elucidate the mechanism through which the digital economy, after incorporating data elements into production factors, influences the centralization of global value chain networks. Once data elements are incorporated into the production function, they significantly boost enterprise productivity. From the perspective of the informatization mechanism, by leveraging the global internet and technologies such as big data, cloud computing, and the Internet of Things, enterprises can swiftly acquire extensive market information, customer demands, and supply chain data, enabling rapid data sharing and processing, thereby enhancing their total factor productivity. Furthermore, in terms of alleviating information asymmetry, the digital economy facilitates efficient information dissemination and sharing through internet platforms, making resource allocation more transparent. Consumers and enterprises can more conveniently access the information they need, reducing transaction costs and improving resource allocation efficiency. Ultimately, when enterprise productivity improves and resource misallocation is mitigated, their position within the global value chain will be elevated.

To provide a more intuitive illustration of the theoretical mechanism framework in this paper, we have further added a diagram of the theoretical mechanism framework in the revised manuscript. It is presented as follows:

Fig. 1 A Diagram Illustrating the Mechanism by Which the Development of the Digital Economy Influences the Centralization of Global Value Chain Network

(2) Strengthen Causality Claims

The study aims to address causality concerns, but there are still implied causal claims in certain sections (e.g., “The development of the digital economy significantly enhances a country's central location in the GVC network”).

While instrumental variables (IV) were used, discussing potential limitations of the IV approach would strengthen credibility.

Response: Thank you for your specific and professional comments and suggestions. This paper investigates the impact of digital economy development on the centralization of global value chain networks. As the review experts have pointed out, causal inference is one of the core elements of this study. Endogeneity, being the most critical issue that causal inference must confront, is addressed in this paper primarily through the instrumental variable method. Two main instrumental variables are employed: First, the number of landline telephones in 1984 for each country is utilized as an instrumental variable for the development of the digital economy; second, we constructed a weighted value combining the digital economy development level of other countries relative to the home country and the average digital economy development level of countries across different income brackets, multiplied by the digitization rate of various industries within each country. This serves as an instrumental variable for the digital economy at the "country-industry" level (for details, please refer to Table 4 and Table 5).

However, we acknowledge that despite addressing endogeneity issues through the instrumental variable method, there is still room for improvement in causal inference. This is an area where we need to make further efforts in the future.

(3) Improve Flow and Readability

Some sections, especially the methodology and theoretical model, are dense with mathematical formulations. Consider adding brief explanations or intuitive interpretations before presenting equations.

The literature review is comprehensive but could be structured better—perhaps by grouping studies into broader themes (e.g., digital economy impact on firms, trade networks, productivity).

Response: Thank you for your specific and professional comments and suggestions. Firstly, in terms of the readability of the article, we conducted a thorough review of the entire text, particularly focusing on the "methodology and theoretical model" sections, which contain a significant number of mathematical formulations. Therefore, before each formula, we provided a brief description of its meaning. For instance, before formula (1), we clarified that it represents a quasi-linear utility function; before formula (19), we explained that it is the baseline regression equation of this paper.

Subsequently, we reorganized the literature review. In accordance with the expert's suggestions, we restructured the literature review section, summarizing it based on the logical progression of the digital economy's impact on firms, trade networks, and productivity. The specific modifications are as follows:

The research related to this paper focuses on the following aspects: Existing research has explored the impact of the digital economy from multiple perspectives, most of which focus on the impact on labor force employment. First, the digital economy represented by AI has a strong substitution effect on the labor force particularly in traditional labor-intensive industries, inevitably leading to an increase in unemployment [9-11]and a decrease in the share of labor [12], but it also has a substitution effect on the traditional labor force in terms of creating new jobs [13]. Second, the digital economy has a significant boost to overall productivity, as it has strong technology-intensive attributes and a strong spillover effect compared to traditional industries, and hence it can have a significant boost to industry-wide productivity [14]. Finally, given the penetration of the digital economy, enterprises face digital transformation, and their production and operation activities are inevitably affected by the digital economy [15,16]. Most of the studies mentioned above have focused on the impact of the digital economy on the "binary margin" of firms' exports, but few studies have focused on its impact on the quality of export products [17]. Given the continuous penetration of the digital economy and the internet, Ma & Hu explored the impact of the digital economy on multiproduct exporters and found that information technology prices have been decreasing, reducing firms' export costs and improving the quality of their export products while also allowing them to spend more resources on producing their core products to maintain their competitiveness in the international market [18]. Additionally, research suggests that information and communication technology (ICT) plays a significant role in driving global value chain (GVC) activities[19]. Studies from both macro and micro perspectives have found that the development of the internet notably promotes GVC trade[20]. This promotion is mainly achieved by enhancing supply-demand matching and fostering communication and collaboration, injecting new vitality into the global value chain[21]. However, there are also opinions suggesting that ICT may have adverse effects on the upgrading of GVCs. Under the global value chain division system driven by labor arbitrage, the substitution effect of automation on labor leads to the contraction of GVC trade[22].

At the same time, there is a large body of literature using social network analysis as a research tool to analyze and measure the intricate trade issues among countries. The advantage of social network analysis is that it can reflect not only the overall trade pattern in the world but it can also reflect the importance and centralization of a country and each industry in the global trade network. If a country is at the center of the trade network, its access to resources and profit is stronger [2]. Traditional social network analysis usually considers each country as a point in the network, and the line linking the points is the bilateral trade flow. The analysis of the global trade network reveals that trade and economic exchange between countries are becoming more frequent [23,24]. Meanwhile, global trade networks are not static. Akerman & Sein studied the change in the global trade network in 1995-2011 and found that it is moving toward intensification, clustering, and decentralization [25]. In their analysis of the global trade network, Amighini & Gorgoni found that developing countries are playing an increasingly important role in global trade and that automated production is having an impact on previous trade patterns [26]. The above studies analyze the change in the global trade network mainly through bilateral trade volumes. However, with the ch

---

## [Decision Letter · Decision Letter 5]

Digital Economy Development and Global Value Chain Network Centralization

PONE-D-23-10130R5

Dear Dr. yue,

We’re pleased to inform you that your manuscript has been judged scientifically suitable for publication and will be formally accepted for publication once it meets all outstanding technical requirements.

Kind regards,

Imran Ur Rahman, Ph.D

Academic Editor

PLOS ONE

Additional Editor Comments (optional):

The authors have completed all the major recommendations provided by the reviewers and editors. I would recommend the authors carefully revise the article for minor errors and check for overstated claims through proofreading. Also confirm and cross-check all the citations and references.

Reviewers' comments:

Reviewer's Responses to Questions

**Comments to the Author**

1. If the authors have adequately addressed your comments raised in a previous round of review and you feel that this manuscript is now acceptable for publication, you may indicate that here to bypass the “Comments to the Author” section, enter your conflict of interest statement in the “Confidential to Editor” section, and submit your "Accept" recommendation.

Reviewer #7: All comments have been addressed

2. Is the manuscript technically sound, and do the data support the conclusions?

Reviewer #7: Yes

3. Has the statistical analysis been performed appropriately and rigorously? 

Reviewer #7: Yes

4. Have the authors made all data underlying the findings in their manuscript fully available?

Reviewer #7: Yes

5. Is the manuscript presented in an intelligible fashion and written in standard English?

Reviewer #7: Yes

6. Review Comments to the Author

Reviewer #7: (No Response)

7. PLOS authors have the option to publish the peer review history of their article (what does this mean? ). If published, this will include your full peer review and any attached files.

**Do you want your identity to be public for this peer review?** For information about this choice, including consent withdrawal, please see our Privacy Policy .

Reviewer #7: No

---

## [Editor Report · Acceptance letter]

PONE-D-23-10130R5

PLOS ONE

Dear Dr. Yue,

I'm pleased to inform you that your manuscript has been deemed suitable for publication in PLOS ONE. Congratulations! Your manuscript is now being handed over to our production team.

Kind regards,

on behalf of

Dr. Imran Ur Rahman

Academic Editor

PLOS ONE